# Particulate emissions from cooking: emission factors, emission dynamics, and mass spectrometric analysis for different cooking methods

Julia Pikmann[1], Frank Drewnick[1], Friederike Fachinger[1], Stephan Borrmann[1,2]

[1]Particle Chemistry Department, Max Planck Institute for Chemistry, Mainz, 55128, Germany
[2]Institute for Atmospheric Physics, Johannes Gutenberg University Mainz, Mainz, 55128, Germany

*Correspondence to*: Frank Drewnick (frank.drewnick@mpic.de)

**Abstract.**

Since most people, especially in developed countries, spend most of their time indoors, they are heavily exposed to indoor aerosols, which can potentially lead to adverse health effects. A major source of indoor aerosols are cooking activities, which release large amounts of particulate emissions (in terms of both number and mass), often with complex compositions. To investigate the characteristics of cooking emissions and what influences these emissions, we conducted a comprehensive study by cooking 19 dishes with different ingredients and cooking methods. The emissions were monitored in real time with several on-line instruments that measured both physical and chemical particle properties as well as trace gas concentrations. The same instrumentation was used to study the influence of cooking emissions on the ambient aerosol load at two German Christmas markets. In contrast to previous studies, which often focus on individual aspects or emission variables, this broad and coherent approach allows a comparison between the influence of different parameters (e.g., ingredients, cooking method, cooking temperature, cooking activities) on the emissions.

We found influence of cooking emissions on six variables: number concentration of smaller (particle diameter $d_p > 5$ nm) and larger ($d_p > 250$ nm) particles, PM (PM$_1$, PM$_{2.5}$, PM$_{10}$), BC, PAH and organic aerosol mass concentration. In general, similar emission characteristics were observed for dishes with the same cooking method, mainly due to similar cooking temperature and use of oil. The temporal dynamics in the emissions of the aforementioned variables, as well as the sizes of the emitted particles, were mainly influenced by the cooking temperature and the activities during cooking. Emissions were quantified using emission factors, with the highest values for grilled dishes, one to two orders of magnitude lower for oil-based cooking (baking, stir-frying, deep-frying), and the lowest for boiled dishes.

For the identification of cooking emissions with the Aerodyne Aerosol Mass Spectrometer (AMS), and more generally for the identification of new AMS markers for individual organic aerosol types, we propose a new plot type that takes into account the mass spectral variability for individual aerosol types. Combining our results and those of previous studies for the quantification of cooking-related organic aerosols with the AMS, we recommend the use of relative ionization efficiency values higher than the default value for organics (RIE$_{Org}$ = 1.4): $2.17 \pm 0.48$ for rapeseed oil-based cooking and $5.16 \pm 0.77$ for soybean oil-based cooking.

## 1 Introduction

Aerosols affect the Earth's climate, air quality, and human health (IPCC, 2021; WHO, 2021). The World Health Organization (WHO) estimates that air pollution causes 6.7 million premature deaths each year, almost half of which are attributable to indoor air pollution (WHO, 2023). People, especially in developed countries, spend a large portion of their time indoors (~90%), and are therefore exposed to indoor aerosol and other pollutants for long periods of time (Diffey, 2011; Goldstein et al., 2021; Liu et al., 2022). When inhaled, the pollutants in the indoor aerosol can cause the formation of radicals that lead to oxidative stress and the

formation of oxygenated species that can induce inflammatory processes (Kreyling et al., 2006). The possible health effects of aerosol exposure are diverse and include respiratory diseases, cardiovascular diseases, allergies, infectious diseases, and cancer (Pope et al., 2004; Pope and Dockery, 2006; Shiraiwa et al., 2017; Xu et al., 2022).

Indoor aerosol composition is influenced by atmospheric infiltration, as well as multiple indoor emission sources (Abbatt and Wang, 2020; Marval and Tronville, 2022). Though a relatively minor source, evaporation and subsequent condensation of substances from furnishings, building materials, and consumer products, can contribute to indoor aerosol mass. The human body itself is a direct and indirect source of aerosols through perspiration, breathing, talking, etc. In addition, various activities in the home (such as cleaning and moving around) lead to the resuspension and emission of aerosol particles. Combustion processes such

as cigarette smoking, candle or wood burning also cause high indoor emissions (Abbatt and Wang, 2020).

Cooking is considered to be one of the most important indoor emission sources, an activity that often occurs on a daily basis in homes as well as on a larger scale, e.g., in restaurants. In a study evaluating personal exposure to indoor aerosol, cooking was identified as the largest contributor to indoor PM (particulate matter) (Zhao et al., 2006). Indoor PM concentrations can increase tremendously depending on cooking activity, with $PM_{2.5}$ peak concentrations (PM of aerodynamic diameter with $d_p < 2.5$ µm) of

up to 1400 µg m$^{-3}$ (Abdullahi et al., 2013). In developing countries, where solid fuels are often used for cooking, the health burden is even higher (Chafe et al., 2014; Martin et al; Nasir and Colbeck, 2013).

Cooking activities also have an impact on ambient aerosol. In urban areas, cooking contributes 5-30% of the organic aerosol in fine particles during typical meal times, as shown by various measurements, including the AMS (Aerosol Mass spectrometer; Crippa et al., 2013; Mohr et al., 2012; Struckmeier et al., 2016), TAG (thermal desorption aerosol gas chromatography−mass

spectrometry; Wang et al., 2020), and filter measurements (Rogge et al., 1991). In mapping measurements near restaurants, performed by Robinson et al. (2018) with an AMS, most of the measured organic aerosol plumes were attributed to cooking emissions with concentrations up to 100 µg m$^{-3}$, demonstrating the potential of cooking emissions to affect local air quality.

During cooking, a large fraction of the emitted particle mass is in the form of fine particles ($PM_{2.5}$), while the particle number concentrations of the emissions are dominated by ultrafine particles ($d_p < 100$ nm). Accordingly, the number and mass size

distributions are dominated by Aitken and accumulation mode particles, respectively (Buonanno et al., 2009; Marval and Tronville, 2022; Wallace et al., 2004; Wallace and Ott, 2011; Yeung and To, 2008). When inhaled, these particles can penetrate deep into the lung to the alveoli. In particular, ultrafine particles can cause stronger reactions or inflammatory processes in the body than larger particles of the same total mass due to their larger specific surface area (Baron et al., 2011; Marval and Tronville, 2022; Thomas, 2013).

Cooking releases a variety of substances, including volatile organic compounds (VOCs) and particulate matter. The major constituents are saturated and unsaturated fatty acids, glycerides, and sugars and their decomposition products, such as levoglucosan. In addition, aromatics, PAHs (polycyclic aromatic hydrocarbons), and aldehydes may be emitted, many of which are hazardous to health (Abdullahi et al., 2013; Cheng et al., 2016; Klein et al., 2016; Liu et al., 2018; Zhao et al., 2007; Zhao et al., 2019).

Studies on individual aspects of emissions from cooking activities have shown that the composition and quantity of emissions are affected by various parameters, such as the cooking method, ingredients, cooking temperature, and the type of fuel used (e.g., Zhang et al., 2010). The particle sizes as well as number and mass concentrations increase with increasing temperature during cooking (Amouei Torkmahalleh et al., 2012; Buonanno et al., 2009; Klein et al., 2016; Zhang et al., 2010). The comparison of different cooking methods such as steaming, boiling, baking, deep-frying, stir-frying, and grilling showed that the lowest emissions

were observed from steaming and boiling, while the highest were observed from grilling, followed by deep-frying and stir-frying (Alves et al., 2015; Lee et al., 2001; Olson and Burke, 2006). The differences are mainly due to the different cooking temperatures

and the use of oil. For example, See and Balasubramanian (2006) measured the particle size distribution of emissions from cooking tofu using five different cooking methods and observed a 24-fold increase in particle number concentration compared to background during frying, compared to a 1.5-fold increase during steaming. Another aspect relevant to the level of particulate

emissions is the smoke point of the oil used. Studies measuring the emissions from heating different oils showed that for oils with high smoke points, such as sunflower and soybean oil, the emissions were 4-9 times lower compared to olive oil with a lower smoke point (Amouei Torkmahalleh et al., 2012; Gao et al., 2013).

In addition to primary aerosol particles, cooking emissions contain substantial amounts of VOCs (e.g., Katragadda et al., 2010; Klein et al., 2016), S/IVOCs (Semi/Intermediate VOCs; Yu et al., 2022), and aldehydes (Takhar et al., 2021), which are potential

precursors for secondary organic aerosol (SOA) formation. SOA production rates from cooking-related gaseous emissions have been determined using oxidation flow reactors that simulate defined intervals of atmospheric aging. These experiments have shown that the amount of SOA from cooking processes compared to the primary aerosol emissions ranges from similar values to more than an order of magnitude higher amounts (Liu et al., 2018; Yu et al., 2022; Zhou et al., 2021) and is strongly dependent on the cooking method (Zhu et al., 2021).

The analysis of cooking emissions is challenging due to the complexity of the emitted mixture, as well as the emission dynamics and concentration variability during cooking. In particular, the ingredients and the cooking method have a strong influence on the emissions (Abdullahi et al., 2013; Marć et al., 2018; Zhang et al., 2010). In addition, the sampling approach itself (e.g., sampling location or dilution of samples) and the analysis procedure (e.g., focusing on peak levels or integrating over the entire cooking process) can have a strong influence on the resulting emission data.

As shown above, there are several studies in the literature that focus on individual aspects of particulate or gas phase emissions from cooking. Few of these studies focus on the emission dynamics during cooking and their dependence on, for example, cooking related activities. Others focus on the physical particle characteristics of the emitted aerosol or on the chemical composition of the emissions. Even within those studies that provide emission factors for different aerosol properties (e.g., particle number or mass), substantial differences in the experimental setup often prevent direct comparability of emission factors obtained in different studies.

To date, there are very few systematic studies that have both investigated the influence of different cooking parameters on the emissions and measured a wide variety of chemical and physical aerosol properties in parallel. Therefore, we conducted a comprehensive study of cooking emissions by performing a series of measurements, cooking 19 dishes with different ingredients and cooking methods. During the cooking, several chemical and physical properties of the emitted primary aerosol were monitored in real time with our mobile laboratory (MoLa, used in stationary measurement mode in the laboratory), including PM, organics

and non-refractory inorganics, BC and PAH mass concentrations, and particle number concentration and size distribution. These on-line measurements allowed the analysis of the emission dynamics during cooking and of the influence of different cooking activities during preparation on the emissions.

The emissions were quantified and emission factors related to the amount of food were determined for all relevant variables. Based on the laboratory measurements, we investigated how the identification of cooking emissions with the AMS and the identification

of new AMS markers in general can be further improved by using a new plot type. Furthermore, the influence of cooking emissions on ambient aerosol was investigated at two German Christmas markets using MoLa. Based on these measurements, the applicability of the laboratory-derived emission factors to ambient data was investigated.

## 2 Methods and instrumentation

### 2.1 Laboratory study design and experimental procedures

For a systematic study, 19 different dishes were cooked in the laboratory (Table 1). The concept was to prepare dishes that are commonly cooked in Central Europe (Germany), including different classes of ingredients and cooking methods, i.e., boiling, stir-frying, deep-frying, baking, and grilling with gas and charcoal. Each dish was prepared to serve approximately four people, and all ingredients were weighed before preparation (Table S1). Rapeseed oil was used in the cooking of all dishes except the boiled dishes, frozen pizza, and brownies. Only salt and pepper were used as condiments unless otherwise noted in Table S1.

**Table 1: List of dishes prepared for the laboratory study (see Table S1 for details).**

| Cooking method | Dishes |
|---|---|
| Boiling | Boiled potatoes, rice, noodles |
| Stir-frying | Fried potatoes, bratwurst, schnitzel, fish, spaghetti Bolognese, stir-fried vegetables, Indian curry |
| Deep-frying | French fries (in pot), French fries (deep fryer), Bavarian doughnut (in pot) |
| Baking | Baked potatoes, frozen pizza, brownies |
| Grilling on gas grill | Steaks, vegetable skewers |
| Grilling on charcoal grill | Steaks |

Each dish was cooked three times on the same day to assess the variability of emissions due to variations in ingredients and cooking process performance between replicates. Background measurements were taken for 20 minutes immediately prior to the start of cooking. Between replicates, we waited for the aerosol concentration to return to a stable background level; if necessary, the room

was ventilated. The cooking process was recorded using a webcam (HD Pro Webcam C920, Logitech, Switzerland) to assign individual concentration changes to activities during cooking. In addition, the surface temperature of the cooked food and cookware was measured repeatedly at selected locations (typically every few min) with an IR thermometer (Fluke 568, Fluke Corporation, USA). During the baking experiments, the temperature of the air inside the oven was continuously monitored using the same thermometer with a thermocouple sensor. The ambient temperature around the kitchen setup was not measured. We estimate that

it ranged from about 18 °C to 25 °C, depending on the outside temperature.

The measurements were performed in an experimental hall with a custom-built kitchen setup consisting of a standard household electric stove with oven (30540 P, Privileg, Germany) and a hood (CH 44060-60 GA, Respekta®, Germany) above it, connected to an exhaust (Fig. 1). The exhaust flow rate $Q_E$ was 7.5 m$^3$ min$^{-1}$. In order to quantitatively capture the cooking emissions, the space between the stove and the hood was enclosed by four Plexiglass walls and only the front glass was left partially open, leaving

a gap of about 50 cm to allow access to the cookware. For the oven and barbecue experiments, additional screens were used to completely capture the emissions. The cooking emissions were sub-sampled from the exhaust pipe above the hood, diluted (1:13) with a dilution system (VKL 10 E, Palas, Germany) using dry, particle-free compressed air (1 bar), and transferred to the instruments inside our mobile laboratory MoLa. Since the dilution with dry air resulted in low relative humidity (< 7 %) we measured dry particles, which may differ in particle size (and thus mass) from particles measured without dilution near the source.

The particle loss within the setup was calculated using the particle loss calculator (von der Weiden et al., 2009) and was found to be negligible for the particle size range relevant to this study.

The stir-fried dishes were prepared in a Teflon-coated frying pan, the boiled dishes in a stainless steel pot, and the deep-fried dishes in a stainless steel pot or a deep fryer (FT 2400.9, 2300 W, 2.5 L oil, Tevion, Germany). A gas and a charcoal grill were used for the barbecue experiments.

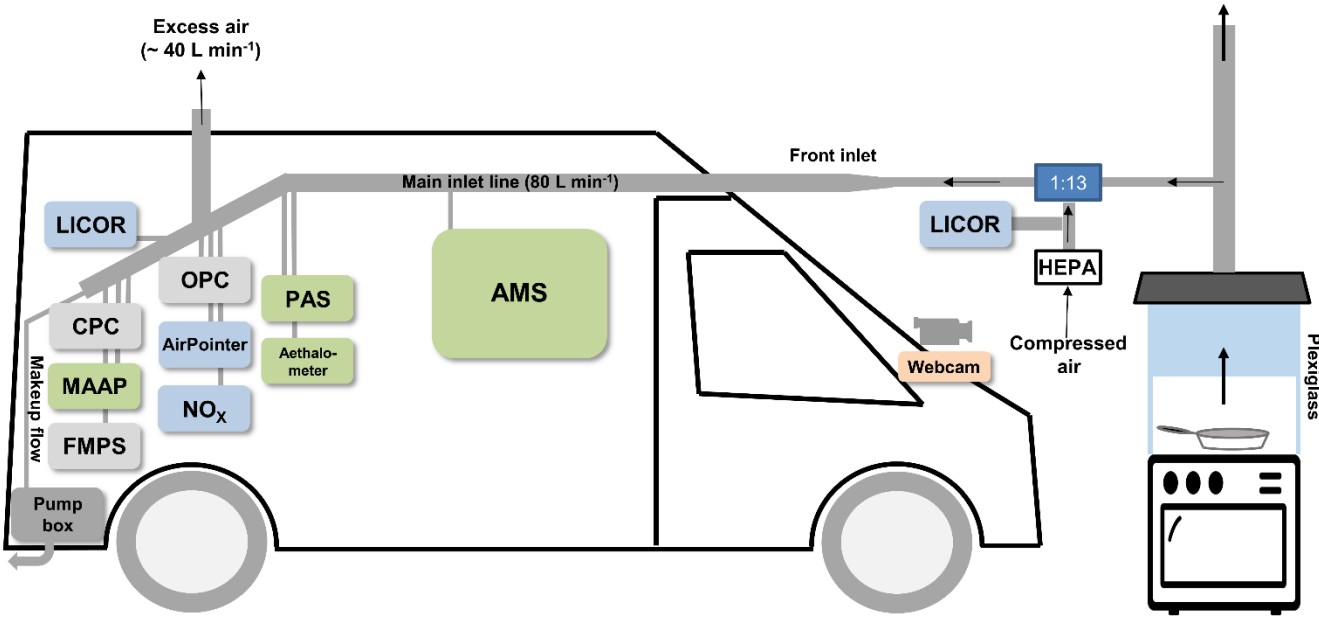


**Figure 1: Scheme of the laboratory setup for the cooking experiments (MoLa scheme adapted from Drewnick et al., 2012). HEPA: high-efficiency particulate air filter. For details on the instrumentation, see Table S2.**

## 2.2 Ambient measurements at two German Christmas markets

Measurements to assess the impact of cooking emissions on local air quality under ambient conditions were performed at two

Christmas markets in Germany:

- Ingelheim (5 to 8 December 2019)

    The Christmas market in Ingelheim (approx. 35000 inhabitants) was located around the Burgkirche; the mobile laboratory MoLa was placed directly behind a circle of seven food stands, offering burgers, French fries, flame-grilled salmon, waffles, vegan food, and mulled wine at the eastern edge of the market. A wood fire barrel was placed 25 meters from

MoLa on 7 and 8 December 2019, and another barrel was placed in the middle of the food stand circle next to MoLa (about 10 meters away) during all evenings. Other food stands and wood-fired barrels were distributed over the market, which covered an area of about 100 m by 50 m. The opening hours were 6 December 5 pm to 10 pm, 7 December 3 pm to 10 pm, and 8 December 3 pm to 9 pm.

- Bingen (13 to 15 December 2019)

The Christmas market in Bingen (approx. 25000 inhabitants) was spread over the city center. MoLa was located at the eastern edge of the Bürgermeister-Neff-Platz, an open area of about 50 m by 25 m, with the nearest food stand at a distance of 25 m. The six food stands on the square were arranged in a semicircle, offering Langos, French fries, bratwurst, barbecue, crepes, raclette, tarte flambée, sweets, and mulled wine. On 14 December, a suckling pig was grilled over an open wood fire on the western edge of the square. On 14 and 15 December, a wood fire barrel was placed in the middle

of the square and another barrel was placed on a crossroad at the western edge of the square. The opening hours were 13 December 4 pm to 9 pm, 14 December 11 am to 9 pm, and 15 December 11 am to 7 pm.

The inlet height for the MoLa instrumentation was 5 m above ground level. We measured mostly dry particles as the elevated temperature inside MoLa led to low relative humidity (< 32 %) in the inlet lines. During the Ingelheim measurements, we additionally measured black carbon mass concentrations with a portable aethalometer (microAeth® MA200, AethLabs, USA)

during random walks through the market.

The temperatures during the measurements at both sites were in the range of 4 – 11 °C and there were occasional light rain showers. The wind direction in Ingelheim was mainly from south-southwest with wind speeds of 1 – 4 m s$^{-1}$ and in Bingen from west with wind speeds of 0.5 – 2 m s$^{-1}$, which resulted in the mobile laboratory being downwind of the Christmas markets most of the time.

## 2.3 Instrumentation

Within the mobile laboratory (MoLa) various instruments were used to measure different aerosol properties such as particle number concentration (measured with a condensation particle counter CPC for particles with $d_p > 5$ nm and with an optical particle counter OPC for particles with $d_p > 250$ nm) and particle size distribution ($d_p = 5.6$ nm – 32 µm, measured with two different instruments: the Fast Mobility Particle Sizer FMPS and the OPC), the mass concentration for the PM$_1$, PM$_{2.5}$, PM$_{10}$ fractions, and the chemical components black carbon (BC) and PAHs in the PM$_1$ fraction as well as the trace gas concentrations of NO$_x$, O$_3$, SO$_2$, CO, and

CO$_2$. The HR-ToF-AMS (high-resolution time-of-flight aerosol mass spectrometer) was used to measure the non-refractory chemical composition of PM$_1$ and was operated in V-mode for maximum sensitivity, with a time resolution of 15 s for the laboratory measurements and 30 s for the Christmas market measurements. An overview of the MoLa instruments, measured variables, time resolutions, and measurement uncertainties is provided in Table S2; for further details on MoLa, see Drewnick et al. (2012).

## 185 2.4 Data processing

All data processing was performed using Igor Pro (versions 6 – 8, WaveMetrics, Inc., USA). Data from the laboratory (Christmas market) measurements were averaged on a common 15 s (30 s) time base. All data were corrected for sampling time delays, checked for invalid data (e.g., due to internal calibrations), and normalized to standard conditions ($T = 20$ °C, $p = 1013.25$ hPa). The sampling dilution (1:13) was taken into account in the further analysis of the cooking experiments. The PM$_1$, PM$_{2.5}$, and PM$_{10}$

mass concentrations were calculated from the combined FMPS and OPC size distribution data (SI Sect. S1). The time-averaged data of individual experiments were averaged over the three replicates (unless otherwise stated), so that the corresponding standard deviation reflects the variability between replicates. For the Christmas market measurements, the open and closed market periods were averaged separately over all days.

To calculate the cooking emissions from the laboratory data, the averaged background concentrations ($c_{Back}$) measured before

each experiment were subtracted from the concentrations measured during cooking ($c_{Cook}$). Identified trends in background concentrations were corrected accordingly. Emission factors ($EF$) were calculated to estimate the total emissions from cooking per kilogram of food according to Eq. (1) from the average concentration of the respective variable ($c_{Avg} = c_{Cook} - c_{Back}$), the exhaust volume flow rate $Q_E$ (7.5 m$^3$ min$^{-1}$), the preparation time $t$, the dilution factor $D$ (13), and the mass of the ingredients $m$.

$$EF = \frac{c_{Avg} \cdot Q_E \cdot t \cdot D}{m} \tag{1}$$

The analysis of the high resolution AMS data was performed with the software tools SQUIRREL 1.63I and PIKA 1.23I within Igor Pro following the standard procedures (Canagaratna et al., 2007). The ionization efficiency of the AMS as well as the relative ionization efficiency for ammonium (4.21) and sulfate (1.31) were determined in calibrations before and after the measurements. For the laboratory data, a collection efficiency (CE) of 1 was applied because we assumed that the emitted particles were mostly composed of liquid components. This assumption is valid only for the laboratory measurements and is based on the observation

that BC and other co-emitted (non-organic) components contribute only about 1% of the total submicron aerosol mass (see Table S6). Using this approach, the relative ionization efficiency for organics (RIE$_{COA}$) was determined for each dish (see Sect. 3.1.4).

For the Christmas market data, the standard values for the CE (0.5) and $RIE_{Org}$ (1.4) were applied (Canagaratna et al., 2007), except for the cooking organic aerosol fraction, as described in Sect. 3.5.1.

For comparison of the measured mass spectra with those of different organic aerosol types from previous studies (Table 2), all
available high-resolution mass spectra of the respective aerosol types were taken from the AMS spectra database (Ulbrich et al., 2009; Ulbrich et al., 2023) as listed in Table S3.

Positive matrix factorization (PMF, Paatero and Tapper, 1994) was performed on the AMS organic high resolution mass spectra up to $m/z$ 116 using the PMF Evaluation Tool (PET) v3.07C (Ulbrich et al., 2009, see SI Sect. S2 for details).

**Table 2: List of organic aerosol types and their acronyms.**

| Acronym | Aerosol type |
|---|---|
| COA | Cooking organic aerosol |
| BBOA | Biomass burning organic aerosol |
| HOA | Hydrocarbon-like organic aerosol |
| OOA | Oxygenated organic aerosol |
| LVOOA | Low-volatile oxygenated organic aerosol |
| SVOOA | Semi-volatile oxygenated organic aerosol |
| LOOOA | Less oxidized oxygenated organic aerosol |
| MOOOA | More oxidized oxygenated organic aerosol |
| NOA | Nitrogen-enriched organic aerosol |
| CCOA | Coal combustion organic aerosol |
| CSOA | Cigarette smoke-related organic aerosol |
| IEPOX-SOA | Isoprene-epoxydiol-derived secondary organic aerosol |

## 3  Results and discussion

### 3.1  Chemical analysis of cooking emissions with HR-ToF-AMS

#### 3.1.1  Average chemical composition and correlation of mass spectra

The mass spectra of non-refractory $PM_1$ cooking emissions from different dishes show a high similarity. On average, the measured
aerosol consisted mainly of organics (96.7 – 99.9 %) with minor contributions from nitrate (< LOD – 2.8 %), ammonium (< LOD – 0.5 %), sulfate (< LOD – 1.8 %), and chloride (< LOD – 0.4 %). Most of the ions in the organic fraction were attributed to the $C_xH_y$ family (77.8 – 91.8 %), indicating a weakly oxidized aerosol. The remaining ions were mostly oxygen containing ions ($C_xH_yO_1$: 6.5 – 17.4 %; $C_xH_yO_{>1}$: < LOD – 6.2 %) with a small fraction attributed to the $C_xH_yN$ family (< LOD – 2.3 %) and the $C_x$ family (0.1 – 0.8 %). For two dishes, Indian curry and spaghetti Bolognese, small fractions of the ions were attributed to the
$C_xS$ family (0.1 %) and the sulfate fraction was also slightly elevated (0.3 – 0.7 %), presumably due to the emission of sulfur-containing substances from onions in the food (Boelens et al., 1971).

To obtain quantitative information on the similarity of emissions from different experiments, linear correlations were calculated between all averaged normalized organic mass spectra (unit mass resolution) of emissions from all dishes (Fig. 2). In addition, the mass spectrum of emissions from heated rapeseed oil (Fig. S1) was included in this comparison. This choice of a comparison
spectrum is based on the fact that rapeseed oil was used in all dishes where oil was required. Most of the spectra show a high degree of similarity to each other and to the spectrum of rapeseed oil (Pearson's $r$ > 0.94), suggesting that the emissions are

associated with oil, which might have vaporized and recondensed. Consistently, the mass spectra of the emissions from boiled dishes and steaks grilled on charcoal are less similar to those of the others: For the boiled dishes, no oil was used, and for the steaks, the mass spectrum is strongly influenced by the emissions from the charcoal itself. In addition, the correlations of the cooking mass spectra with those of different fatty acids (palmitic, stearic, oleic, and linoleic acid), all measured by AMS (Ulbrich et al., 2023, not shown in Fig. 2), show the highest similarity with that of oleic acid ($r = 0.85 - 0.94$), the main component of rapeseed oil and many other cooking oils. These observations suggest that a substantial fraction of cooking-related emissions are fatty acids, either from the used cooking oils or from components of the prepared food. This is consistent with the fact that oil components may vaporize and recondense, and fats contained in the food may produce condensable fatty acids after decomposition. In contrast, peptides and carbohydrates are more likely to decompose into products that either remain in the gas phase or do not vaporize under the cooking conditions.

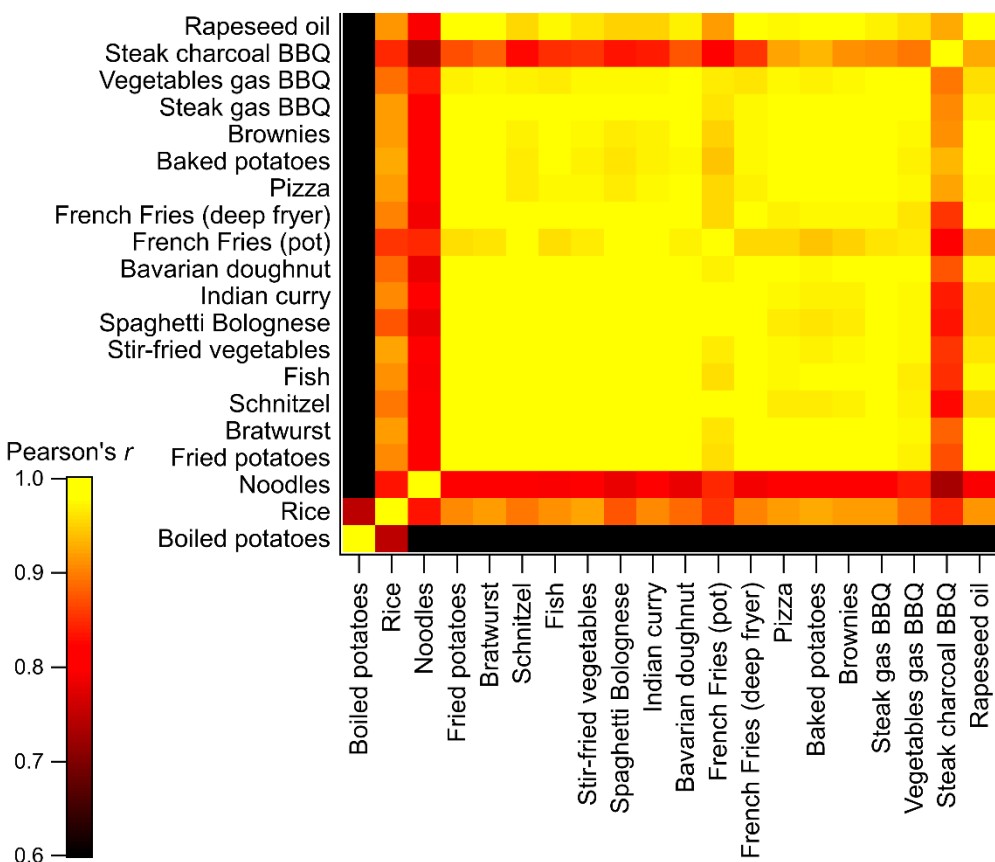

**Figure 2: Linear correlation of the averaged mass spectra of cooking emissions for all laboratory experiments and pure rapeseed oil, color-coded based on the respective correlation coefficient (Pearson's $r$).**

Furthermore, correlations of the cooking mass spectra from this study with mass spectra of different organic aerosol types from previous studies were calculated (Fig. S2). The latter, obtained by PMF analysis of field measurement data, were taken from the AMS spectra database (Ulbrich et al., 2023) and averaged over all available spectra for the respective aerosol type (see Table S3 for the list of mass spectra used). The highest similarity of mass spectra related to oil- or fat-containing dishes was observed with the average COA mass spectrum ($r = 0.92 - 0.98$); therefore, we conclude that also during field measurements, the mass spectra of cooking-related emissions are substantially influenced by the mass spectral patterns of vaporized and recondensed oil or fatty acids. Furthermore, a strong correlation was observed between the mass spectra of the steak over charcoal grilling experiment and that of HOA, presumably due to the contribution of charcoal combustion to the total emissions in this case.

### 3.1.2   Characteristics of mass spectra from cooking emissions

The main characteristics of the mass spectra from the cooking experiments are shown in Fig. 3 for the "frying bratwurst"

experiment as an example. The highest signal intensities were found at *m/z* 41 and 55, except for the boiled dish experiments. These signals are due to emissions of unsaturated hydrocarbons, presumably unsaturated fatty acids (He et al., 2010; Mohr et al., 2009). The most prominent ion series in the mass spectra are $C_nH_{2n+1}^+$ and $C_mH_{2m+1}CO^+$ (*m/z* 29, 43, 57, 71, …), and $C_nH_{2n-1}^+$ and $C_mH_{2m-1}CO^+$ (*m/z* 41, 55, 69, 83, …) from alkanes, alkenes, and oxygenated substances such as acids, especially fatty acids. In addition, the ion series $C_nH_{2n-3}^+$ (*m/z* 67, 81, 95, 107, …) and $C_6H_5C_nH_{2n}^+$ (*m/z* 77, 91, 105) indicate the presence of cycloalkanes

and aromatic hydrocarbons (Alfarra et al., 2004; He et al., 2010; McLafferty and Turecek, 1993; Mohr et al., 2009).

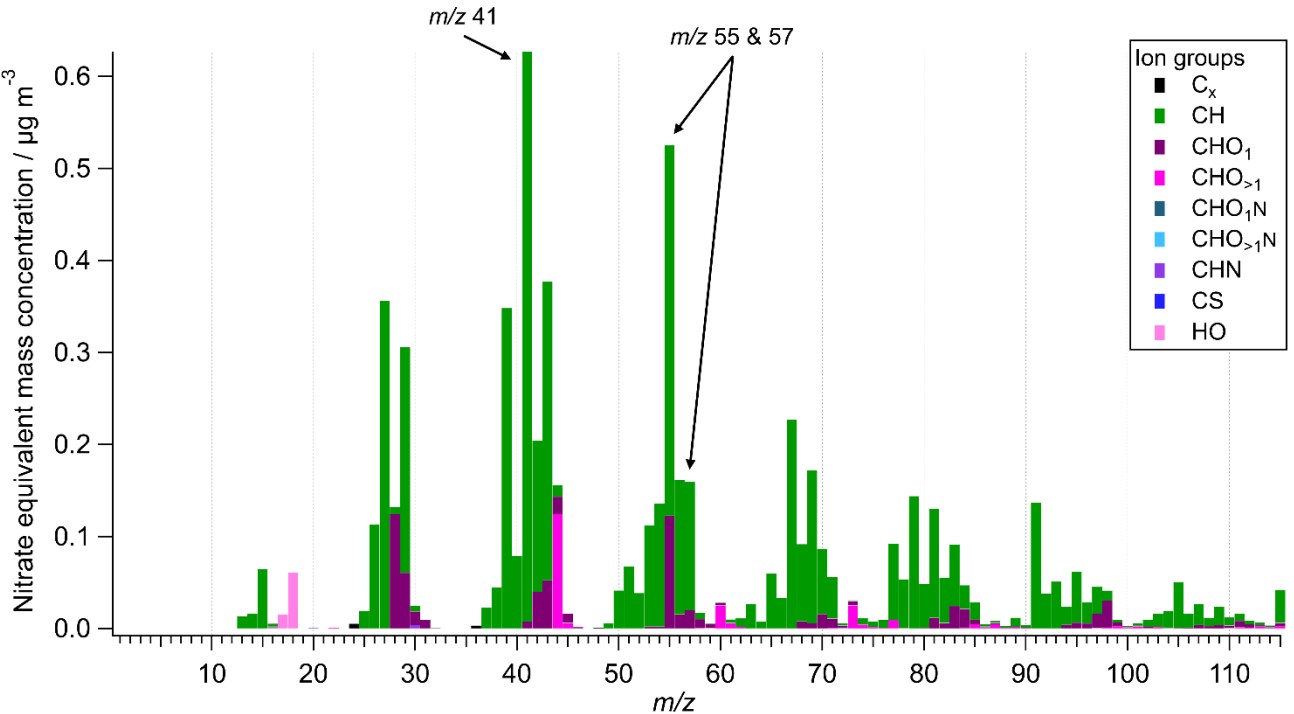

**Figure 3: Unit-resolution mass spectrum of organic aerosol emitted from frying bratwurst with important *m/z* marked.**

An indicator for COA is a high ratio of *m/z* 55 to *m/z* 57 in the mass spectra. In our experiments, the observed ratio was 2.3 − 4.5,

except for the boiled potatoes and the charcoal grilled steaks with 1.3 and 1.7, presumably due to the fact that the respective emissions are not dominated by vaporized oil and decomposed fats. This is in good agreement with the observation of ratios of typically above two for COA-related mass spectra (Mohr et al., 2012; Sun et al., 2011; Xu et al., 2020).

As another meaningful marker for cooking-related organic aerosol we identified the ratio of *m/z* 67 to *m/z* 69 in the spectra. For our cooking experiments, this ratio was in the range of 1.1 – 1.6, again excluding the boiled potatoes and the charcoal grilled steaks

experiments with 0.81 and 0.7, respectively. The ratio for COA obtained from previous PMF analyses of ambient measurements is 1.2 ± 0.1 (Ulbrich et al., 2023), while different results have been obtained from direct measurements of cooking aerosols. For emissions from Chinese cooking, heating sunflower, soybean, corn, and rapeseed oil, and frying sausages and French fries with rapeseed and sunflower oil, the ratio was above 1 (Faber et al., 2013; He et al., 2010; Liu et al., 2017a; Liu et al., 2017b; Xu et al., 2020), while Allan et al. (2010) and Zhang et al. (2021) measured ratios below 1 for heating or cooking with rapeseed, sunflower,

peanut, and corn oils; it was also below or close to 1 for barbecue emissions, frying meat, heating olive and palm oils, and lard

(Kaltsonoudis et al., 2017; Liu et al., 2018; Xu et al., 2020). For HOA, BBOA, LVOOA, and SVOOA, ratios ranging from 0.63 to 0.88 were observed (Table S4).

Considering these studies, we conclude that the ratio of $m/z$ 67 to $m/z$ 69 in the mass spectra depends on the fatty acid composition and the fraction of polyunsaturated fatty acids in the measured aerosol. For saturated and monounsaturated fatty acids the ion series

$C_nH_{2n-1}^+$ and $C_mH_{2m-1}CO^+$ ($m/z$ 41, 55, **69**, 83, …) are more prominent, while for polyunsaturated fatty acids the ion series $C_nH_{2n-3}^+$ ($m/z$ **67**, 81, 95, 107, …) is dominant (Christie 2023; Hallgren et al., 1959). For rapeseed, sunflower, and corn oils the polyunsaturated fatty acid fraction is above 25% and the ratio of $m/z$ 67 to $m/z$ 69 is mostly above 1. For oils with lower fractions of polyunsaturated fatty acids, such as palm or olive oil, and animal fats, such as lard, the ratio is below 1. Thus, the ratio of $m/z$ 67 to $m/z$ 69 could be an indicator of the composition of the oil used for cooking.


During the laboratory cooking experiments, increased signal intensities were observed for $m/z$ 60 and 73. We also found these enhanced signal intensities for emissions from pure heated rapeseed oil and they were also observed in reference mass spectra of the fatty acids oleic, stearic, and palmitic acid (AMS spectra database, Ulbrich et al.2023). Frequently, high signal intensities at $m/z$ 60 and 73 in AMS mass spectra are indicative of biomass burning aerosol due to the fragments $C_2H_4O_2^+$ and $C_3H_5O_2^+$ of

levoglucosan generated by pyrolysis of cellulose (Schneider et al., 2006). However, elevated signal intensities at these $m/z$ in cooking-related aerosols are likely to originate from fatty acids rather than from levoglucosan, i.e. the ion structure contains a carboxyl group rather than a diol (Fachinger et al., 2017), leading to a different fragmentation pattern. Thus, one possibility to differentiate between biomass burning and cooking emissions could be the ratio of $m/z$ 60 to 73. The ratios for pure levoglucosan and BBOA are 3.7 and 1.5, respectively, while the ratios from the cooking experiments, excluding the boiled dishes due to low

organic concentrations and high uncertainty, ambient COA, and fatty acids are at most 1.1 (Table 3). Similar observations were reported by Xu et al. (2020) who measured a ratio of ~2 for BBOA and around 1 for COA. However, since the ratio of $m/z$ 60 to 73 for HOA (Table 3) is not significantly different from those of the various COA-related values, it cannot be used by itself to discriminate between these two types of organic aerosols (see also Fig. S3).


**Table 3: Ratio of signal intensities at *m/z* 60 and 73 from mass spectra of different compounds and aerosol types. For BBOA, HOA, COA, and the cooking experiments, the average and standard deviation were calculated from the available data. All mass spectra except for the cooking experiments were obtained from the AMS spectra database (Ulbrich et al., 2023).**

| | Ratio of signal intensities at *m/z* 60 and 73 |
|---|---|
| **BBOA-related** | |
| Levoglucosan | 3.71 |
| BBOA | $1.47 \pm 0.53$ |
| **HOA-related** | |
| HOA | $0.95 \pm 1.12$ |
| **COA-related** | |
| Oleic acid | 0.81 |
| Stearic acid | 0.87 |
| Palmitic acid | 0.89 |
| COA | $1.10 \pm 0.13$ |
| Cooking experiments (our study) [a] | $0.90 \pm 0.08$ |
| Rapeseed oil | 0.95 |

[a]excluding boiled dishes (low organic concentrations)

### 3.1.3 Discrimination of different aerosol types based on markers in their mass spectra

Ambient aerosol is usually a mixture of different aerosol types due to the contribution of different aerosol sources and aging processes in the atmosphere. In order to identify the individual aerosol types and their contribution to the total aerosol, PMF is applied to the mass spectra of the measured organic aerosol fraction and the factors obtained are attributed to the different aerosol types using different indicators and by comparison with other available data. For this study, a new plot type was used to assess whether combinations of known and new indicators in the mass spectra are suitable to reliably discriminate between different aerosol types and to check whether PMF works well to separate different aerosol contributions. While in some cases (like $f60/f73$, see Sect. 3.1.2) individual markers might be sufficient to reasonably differentiate between different aerosol types, using such a combination of indicators can give a more robust information also in cases where differences between individual markers are less pronounced between different aerosol types.

In these "rectangle plots" the values of two indicators for all available aerosol types are plotted against each other in an xy-plot. The standard deviation or uncertainty for each indicator of a particular aerosol type is reflected in the x- and y-directions by a box to show the variability of the mass spectra for that aerosol type. The different aerosol types are well separated with a selected combination of indicators if the boxes do not overlap. Indicators for individual aerosol types can be the fraction of the signal intensity at a single *m/z* out of the total organic signal, e.g. $f_{44}$ for the signal fraction at *m/z* 44, a combination of such fractions, e.g. $f_{55}/f_{57}$, or elemental ratios of the organic aerosol, such as $O/C$ and $H/C$.

For the cooking experiments, the respective values were calculated as the average of the three replicates, while for the individual reference aerosol types the available mass spectra from the AMS spectra database (Ulbrich et al., 2023) were averaged. In both cases the corresponding standard deviation is shown as a box; if only one reference mass spectrum or one replicate was available (e.g., rapeseed oil, RO), the variability observed during the respective measurement is used as the uncertainty in the rectangle plot.

The boiled dishes experiments and one of the deep-frying French fries experiments, in which the frying oil cooled down strongly due to too many French fries used, were excluded from this analysis due to very low organic concentrations and resulting high uncertainties.

Plotting the two known COA markers, $f_{55}$ and $f_{55}/f_{57}$, together in such a rectangle plot (Fig. 4) shows that the mass spectra of ambient COA and from the cooking experiments are well separated from those of other aerosol types with this combination of

markers. The values for COA and the cooking experiments are located in the upper right corner with high $f_{55}$ (> 0.06) and $f_{55}/f_{57}$ (> 2) values. Although the COA and HOA mass spectra are often similar, when both markers are used in combination, they are well separated from each other, except for the charcoal grilled steaks experiment, which is located within the HOA box. The $f_{55}$ values for the cooking experiments are slightly lower than those of the ambient COA while the $f_{55}/f_{57}$ values are similar for both or slightly higher than those of the ambient COA. This could either be due to the difference between ambient and laboratory aerosol, as

ambient aerosol can chemically change in the atmosphere, or because PMF is not able to completely separate the different aerosol types; it could also simply reflect the fact that the cooking experiments represent single source processes, whereas the ambient COA data represent an aerosol that is a mixture of a large number of sources. The PMF results from the Christmas market measurements (Sect. 3.5) can be found in the same area of the rectangle plot, but partially outside the one-sigma range of the literature COA results. It is noteworthy that the box representing the results of the pure rapeseed oil measurements is shifted to

slightly larger $f_{55}$ values compared to those from the laboratory cooking experiments. This result suggests that although the correlation analysis shows a high similarity between the rapeseed oil and the cooking emission mass spectra, the cooking emissions contain other components in addition to rapeseed oil. Similarly, for pure oleic acid (taken from the AMS spectra database), $f_{55}$ is significantly larger than the value found for the cooking related aerosols and rapeseed oil, probably due to the fact that the latter also contain other components. The rectangle plot of $f_{55}/f_{57}$ versus $f_{55}$ also shows that, based on this combination of COA markers

for example, BBOA is not well separated from CCOA, CSOA and several OOA aerosol types.

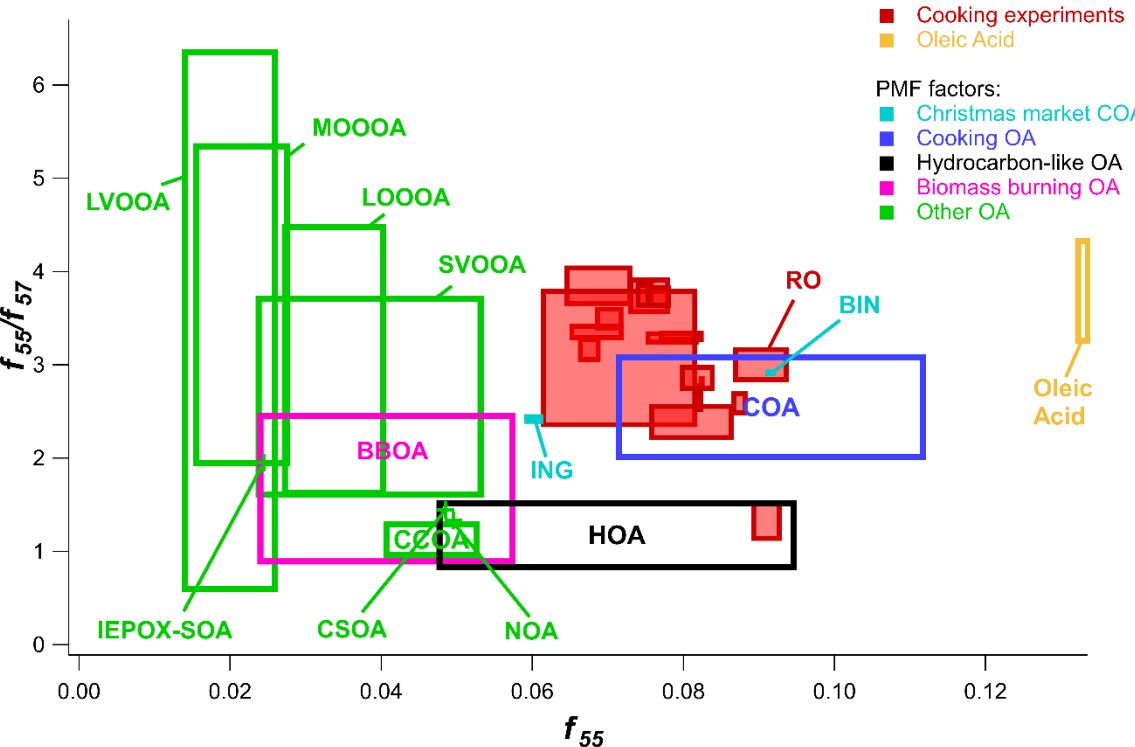

**Figure 4: "Rectangle plot" of $f_{55}/f_{57}$ combined with $f_{55}$ for the cooking experiments and different organic aerosol types from ambient measurements. The rectangles represent one standard deviation of the markers for the respective aerosol types as found in mass spectra**

**from the literature. The acronyms for the different aerosol types are listed in Table 2; RO stands for rapeseed oil; ING and BIN stand for the Christmas market measurements in Ingelheim and Bingen, respectively. The rectangles representing the cooking experiment results are shaded for better distinction.**

To determine whether the $f_{67}/f_{69}$ ratio is suitable as a COA marker, these ratios were plotted together with $f_{55}$ in another "rectangle plot" (Fig. 5). Also for this combination of markers, the cooking results are found in the upper right area of the plot, well separated

from the other aerosol types. Only for the charcoal grilled steak experiment was an overlap found with the HOA box. The rapeseed oil results are found on the higher $f_{55}$ side of the laboratory cooking experiments, as in the previous rectangle plot (Fig. 4). From these results we conclude that the $f_{67}/f_{69}$ ratio may be a marker for COA similar to the $f_{55}/f_{57}$ ratio, but the influence of the fatty acid composition of the emitted oil or fat must be considered (see Sect. 3.1.2). Therefore, the $f_{67}/f_{69}$ ratio should only be used as an additional marker for COA.

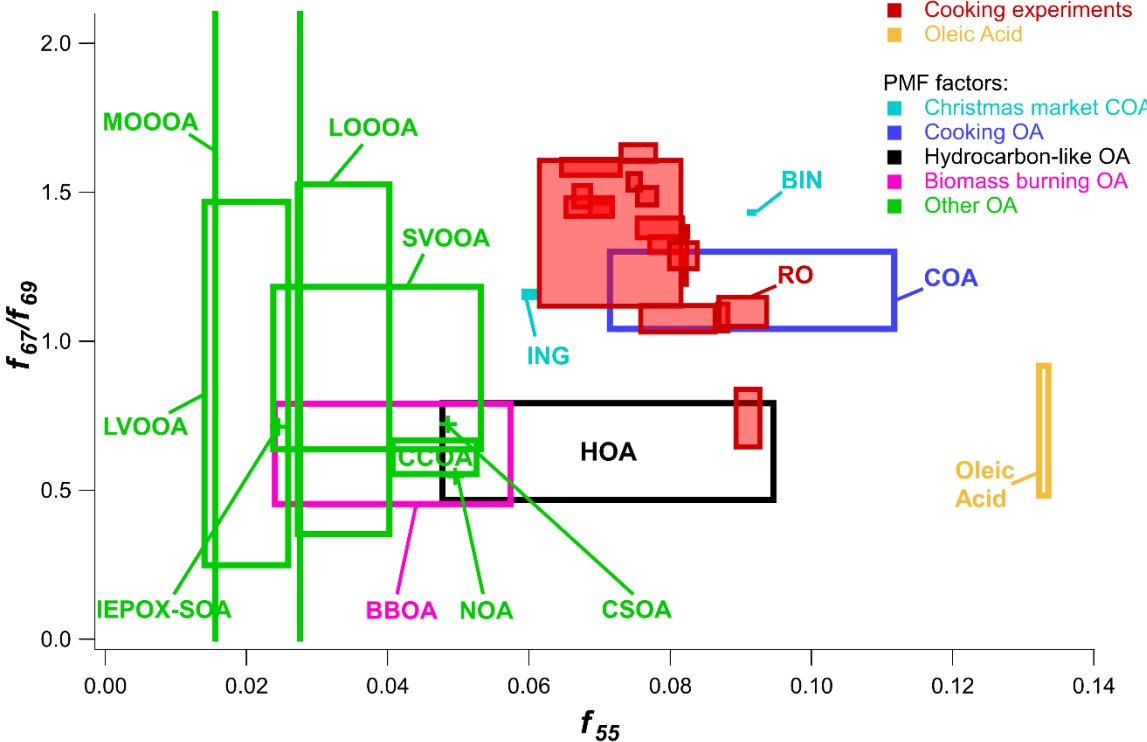


**Figure 5: "Rectangle plot" of $f_{67}/f_{69}$ combined with $f_{55}$ for the cooking experiments and different organic aerosol types from ambient measurements. The rectangles represent one standard deviation of the markers for the respective aerosol types as found in mass spectra from the literature. The acronyms for the different aerosol types are listed in Table 2; RO stands for rapeseed oil; ING and BIN stand for the Christmas market measurements in Ingelheim and Bingen, respectively. The rectangles representing the cooking experiment**
**results are shaded for better distinction.**

The high-resolution mass spectra from the cooking experiments were used to extract further information about the individual ions that contribute to the specific cooking-related mass spectra. Due to the strong fragmentation of organic molecules in the AMS analysis process, the individual ions measured by the instrument provide little information about the corresponding aerosol components. For this reason, AMS organics analysis typically reports ion families (i.e., groups of ions containing specific

combinations of atomic contributions) rather than individual ions. For the $m/z$ discussed in the previous "rectangle plots" ($m/z$ 55, 57, 67, and 69), ions associated with the $C_yH_y$ and $C_xH_yO$ ion families are observed for the cooking experiments. In addition, ions of the $C_xH_yO_2$ family are found at very low abundance in the $m/z$ 57 and $m/z$ 69 signals.

Table 4 illustrates the contribution of different ion families to the individual marker $m/z$ signals and corresponding ions. For each ion family at each $m/z$, in addition to the main ion, an isotope ion containing $^{13}C$ is listed. These contribute approximately 2-3% of

the respective family signal. For all marker $m/z$, the signal is dominated by the pure hydrocarbon ions (i.e., the ions from the $C_xH_y$ family) with smaller relative contributions for $m/z$ 55 and 57 (75% and 86%, respectively), in comparison to those for $m/z$ 67 and 69 (~100% and 96%, respectively). Consequently, the relative contribution of oxygen-containing ions is larger for $m/z$ 55 and 57 and almost negligible for $m/z$ 67. The uncertainties provided in Table 4 are the standard deviations for the individual relative ion

family contributions, calculated from all cooking experiments. The uncertainty due to background subtraction and variations in background concentrations is much smaller than the variability between individual cooking experiments and is included in these values. In general, no significant difference in the relative contributions of the different ion families is observed across different cooking methods, with the exception of grilling, which shows a notable difference in the $m/z$ 57 and $m/z$ 69 composition. The contribution of the $C_xH_yO$ family ions in the grilling experiments is significantly higher (18.1% for $m/z$ 57 and 6.4% for $m/z$ 69) than in the other experiments (12.4% for $m/z$ 57 and 3.5% for $m/z$ 69). This suggests that the grilling method results in an enhanced production of oxygen-containing substances, in comparison to the other cooking methods.

**Table 4: Contribution of individual ion families and their associated ions to the ion signal at the four cooking-related marker $m/z$.**

| $m/z$ | $C_xH_y$ family | | $C_xH_yO$ family | | $C_xH_yO_2$ family | |
|---|---|---|---|---|---|---|
| | ions | contribution | ions | contribution | ions | contribution |
| 55 | $^{13}CC_3H_6^+$, $C_4H_7^+$ | 75±4% | $^{13}CC_2H_2O^+$, $C_3H_3O^+$ | 25±4% | - | - |
| 57 | $^{13}CC_3H_8^+$, $C_4H_9^+$ | 86±5% | $^{13}CC_2H_4O^+$, $C_3H_5O^+$ | 14±5% | $^{13}CCO_2^+$, $C_2HO_2^+$ | 0.3±0.4% |
| 67 | $^{13}CC_4H_6^+$, $C_5H_7^+$ | 99.9±0.3% | $^{13}CC_3H_2O^+$, $C_3H_3O^+$ | 0.1±0.3% | - | - |
| 69 | $^{13}CC_4H_8^+$, $C_5H_9^+$ | 96±2% | $^{13}CC_3H_4O^+$, $C_4H_5O^+$ | 4±2% | $^{13}CC_2O_2^+$, $C_3HO_2^+$ | 0.3±0.3% |

### 3.1.4 Relative ionization efficiency of cooking-related organic aerosol

The quantification of the aerosol species measured with the AMS is based on Eq. (2) (Canagaratna et al., 2007)

$$C_S = \frac{10^{12} \, MW_{NO_3}}{CE_S \, RIE_S \, IE_{NO_3} \, Q_{AMS} \, N_A} \sum_{all \ i} I_{S.i} \tag{2}$$

where the ion rates of species $S$, $I_{S,i}$, summed over all $i$ $m/z$, are converted to mass concentrations $C_S$, with $MW_{NO_3}$ the molecular weight of nitrate (in g mol$^{-1}$), $Q_{AMS}$ the volumetric inlet flow rate (in cm$^3$ s$^{-1}$), $N_A$ Avogadro's number and $10^{12}$ a unit conversion factor to µg m$^{-3}$. The remaining (unitless) factors in Eq. (2) are from calibrations or based on assumptions. The collection efficiency $CE_S$ for the species $S$ is the ratio of the particle mass measured by the AMS to the particle mass introduced into the inlet. It is mainly influenced by the particle phase, solid or liquid. The typical value for ambient aerosol is 0.5, which accounts for mainly solid particles, a fraction of which bounces off the vaporizer without being vaporized. For the particles from the presented cooking experiments, a CE value of 1 was chosen, assuming that the emitted aerosol contained substantial amounts of liquid oil (see Sect. 3.1.1), which suppresses bounce (Matthew et al., 2008).

The ionization efficiency of nitrate $IE_{NO_3}$, determined in a calibration, is used as the basis for calculating the ionization efficiencies for other species, using the relative ionization efficiency of species $S$ ($RIE_S$) relative to $IE_{NO_3}$. The default value for RIE$_{Org}$ is 1.4, based on laboratory experiments with different types of organic species (Canagaratna et al., 2007). Because COA concentrations measured with the AMS in previous studies were found to be higher than those from parallel measurements with other instruments, RIE$_{COA}$ is assumed to be greater than 1.4 (Katz et al., 2021; Reyes-Villegas et al., 2018; Yin et al., 2015).

In this work, RIE$_{COA}$ was determined by comparing the PM$_1$ mass concentration determined from the FMPS and OPC measurements (PM$_1$) with the total AMS and black carbon mass concentration (PM$_{1,AMS+BC}$) measured in parallel. The oven and boiling experiments were excluded from this analysis due to almost exclusively low measured organic mass concentrations (< 1 µg m$^{-3}$). The density of the fine particles used to calculate PM$_1$ from the particle volume was in the range of 0.91 – 1.03 g cm$^{-3}$ (Table S5), determined individually for each dish (see Sect. S1). These values are in good agreement with the densities for cooking emissions found by Katz et al. (2021) (0.95 – 1.0 g cm$^{-3}$), and, considering their uncertainty of 15%, also with that of rapeseed oil

(0.91 g cm$^{-3}$), consistent with our assumption that the particulate emissions from the cooking experiments contained substantial amounts of vaporized and recondensed oil or fatty acids (see Sect. 3.1.1).

The measured PM$_1$ was mostly composed of organics (see Sect. 3.1.1; the contribution of BC was negligible); consequently, as expected, PM$_{1,AMS+BC}$ was higher for most of the cooking experiments compared to PM$_1$ when using the default RIE$_{Org}$ = 1.4. To determine the RIE$_{COA}$ for each experiment (or, more precisely, the product of RIE$_{COA}$ and CE; we assume CE = 1), the PM$_{1,AMS+BC}$

time series was correlated with that of PM$_1$ for each experiment separately and the RIE$_{COA}$ was adjusted to obtain a slope of 1 for the correlation. For the grilling experiments, the RIE values were determined separately for the "grilling" and "grill warm-up" experimental phases, the latter not being considered as RIE$_{COA}$. A typical example correlation for each cooking method is shown in Figure S4. The resulting RIE$_{COA}$ values for the cooking experiments were in the range of 1.53 – 2.52 and thus frequently substantially higher than the default value of 1.4 (Fig. 6 and Table S5). The uncertainty for the determined RIE$_{COA}$ value was

estimated to be 38%, based on the method of Katz et al. (2021) with uncertainty propagation (see Sect. S3).

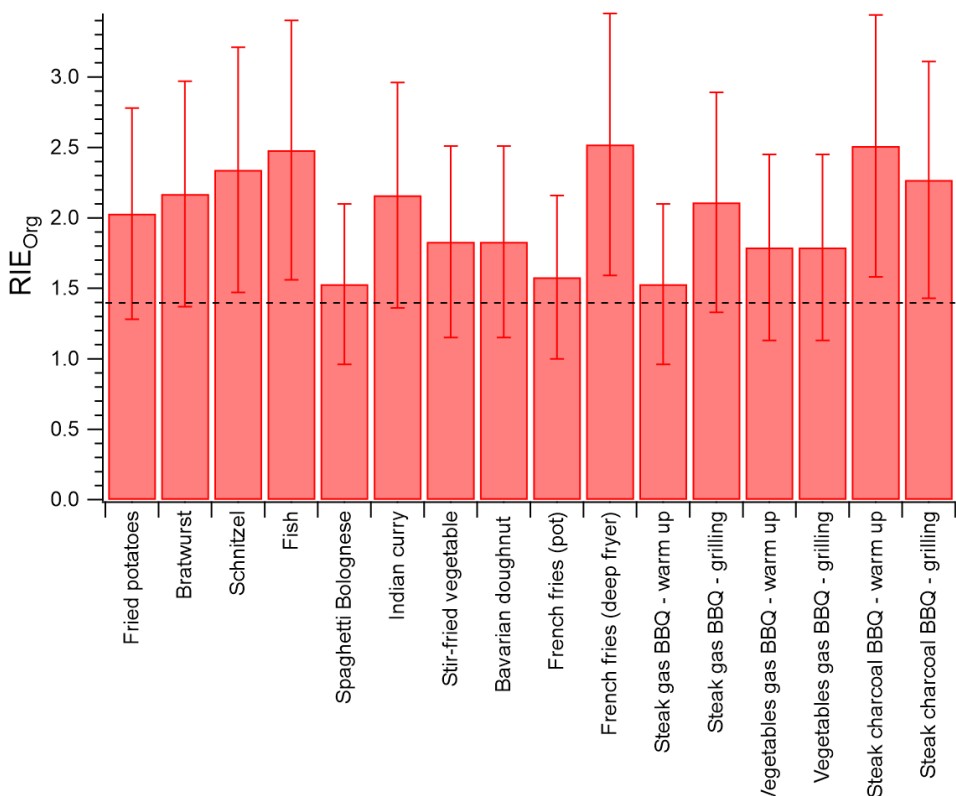

**Figure 6: RIE$_{COA}$ obtained for the different cooking experiments. The default RIE$_{Org}$ of 1.4 is shown as a dashed line.**

In previous AMS studies of cooking emissions, the RIE$_{COA}$ determined was also greater than 1.4. Reyes-Villegas et al. (2018)

determined RIE values of 1.56 – 3.06 for cooking emissions from different types of dishes, comparable to our results, by comparing the measured concentrations (CE = 1) with scanning mobility particle sizer (SMPS) size distribution measurements ($d_p$ = 18 – 514 nm). In contrast, Katz et al. (2021) found substantially higher RIE$_{COA}$ values of 4.26 – 6.50 from indoor aerosol measurements during cooking experiments with CE = 1, also by comparison with SMPS data ($d_p$ = 4 – 532 nm). A possible explanation for the higher values of Katz et al. (2021) could be that the RIE$_{COA}$ depends on the fatty acid composition of the oil or fat containing

droplets. For oleic acid, the main fatty acid of rapeseed oil used in the present study and that of Reyes-Villegas et al. (2018), Katz et al. (2021) obtained an RIE value of 3.18 ± 0.95, similar to the value of 3.0 measured by Xu et al. (2018), while for linoleic acid,

the main component of soybean oil, used by Katz et al. (2021) for their cooking experiments, an RIE value of 5.77 ± 1.73 was found.

Summarizing the results of the current and previous studies, we recommend an $RIE_{COA}$ greater than 1.4 for the COA fraction of the measured organic aerosol for measurements near cooking emission sources. Depending on the cooking oil, which is expected to have a strong influence on the $RIE_{COA}$ value, we suggest an average $RIE_{COA}$ of 5.16 ± 0.77 (average of all measurements with standard deviation) for soybean oil-based cooking, based on the measurements of Katz et al. (2021), while for rapeseed oil-based cooking we recommend an average $RIE_{COA}$ of 2.17 ± 0.48 (average of the averages of both studies with standard error), based on the measurements presented in this study and those of Reyes-Villegas et al. (2018). The individual values used for this estimate are listed in Table S5.

## 3.2 Emission dynamics related to temperature and cooking activities

In order to study the emission dynamics during cooking as a consequence of different activities, the concentration time series obtained for all dishes and for all measured variables were examined in combination with the webcam recordings. For six emission variables, increases and changes over the cooking time were identified: particle number concentration of smaller and larger particles measured by CPC (PNC, $d_p > 5$ nm) and OPC ($PNC_{d>250\ nm}$), PM concentration ($PM_1$, $PM_{2.5}$, $PM_{10}$), BC, PAH, and organics mass concentrations (shown in Fig. S5 as an example for the "frying bratwurst" experiment). Of these six, $PNC_{d>250nm}$, organics and PM mass concentrations are all related to the total emitted particle mass and therefore show similar emission dynamics. No increase above the detection limit was observed for the measured trace gas concentrations, except for $NO_x$ during the grilling experiments and $SO_2$ during the charcoal grilling experiment.

Two types of systematic changes were observed for the six variables. First, the measured concentrations for these variables increased over the cooking time, along with a general increase in food and cookware temperature, as deduced from repeated manual temperature measurements with the IR camera. The emission concentrations usually started to increase only after a certain heating or cooking time, probably when the used oil and the food reached a certain temperature. Also, during sufficiently long periods of inactivity, i.e. more than about 30 – 60 s, the $PNC_{d>250\ nm}$ and organics mass concentrations increased, probably because certain parts of the food reached sufficiently high temperatures. Such increased particle mass and number emissions with higher temperature were also observed in previous studies, e.g. by Buonanno et al. (2009), Amouei Torkmahalleh et al. (2012), and Zhang et al. (2010).

The reason for this progressive increase in concentration is presumably the increased vaporization of substances with increasing temperatures. After emission, the vaporized substances cool down again, eventually leading to increased particle number and mass concentration due to nucleation and recondensation. Correspondingly, the emission concentrations decreased as the power of the stove was reduced.

An increase in BC and PAH mass concentrations has only been observed for high-temperature cooking methods such as grilling or the final stage of stir-frying. PAHs are formed at high temperatures, especially above 400 °C, and due to incomplete combustion, such as during grilling, where BC is also formed (Jägerstad and Skog, 2005; Lijinsky, 1991; Omidvarborna et al., 2015). The described dependence of the measured concentrations on temperature is schematically illustrated in Fig. 7. Due to the substantial heterogeneity of the temperature distribution throughout the food and cookware and the unknown location of the generation of emissions, this relationship can only be presented qualitatively.

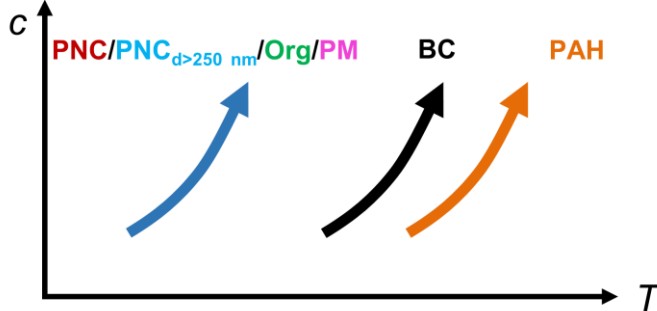

**Figure 7: Schematic diagram of the temperature (*T*) dependence of the emission concentrations (*c*) of six relevant species.**

The second systematic observation is the short-term concentration changes associated with various activities during cooking, such as tilting the pan or turning the food, which have not been studied in such detail before. The activities leading to these short-term changes are shown schematically in Fig. 8, grouped by emission variable, with the increase factors by which the concentrations change from just before the increase to the corresponding maximum concentration. The factors are color-coded: green for relative increases of less than one order of magnitude, yellow for increases of more than one order of magnitude, and red for increases of

more than two orders of magnitude.

Presumably the emission concentrations increase briefly when hot material from the cooked food is brought to the surface by stirring or similar activities, facilitating vaporization. This leads to increased particle formation and growth through condensation of these substances. In addition, contact between cold, water-containing foods and highly heated surfaces, such as the pan, grill, or hot oil, results in rapid vaporization of oil, various other substances, and especially water, which can lead to bubbling of the oil.

The resulting increase in oil surface area presumably leads to increased vaporization of oil and mechanical formation of larger particles due to the bursting of oil bubbles. These processes decrease rapidly as the hot surface cools. Similarly, momentary increases in concentration occur when droplets or components of the grilled food, as well as residues from cleaning the grate, fall onto hot surfaces, such as the charcoal, and quickly vaporize or burn. The high temperatures at these locations also cause transient increases in BC and PAH concentrations. The largest relative increases in emission concentrations for almost all variables were

observed when the oven was opened during baking, presumably due to the low concentrations before the oven was opened and the sudden release of emissions that had accumulated in the oven.

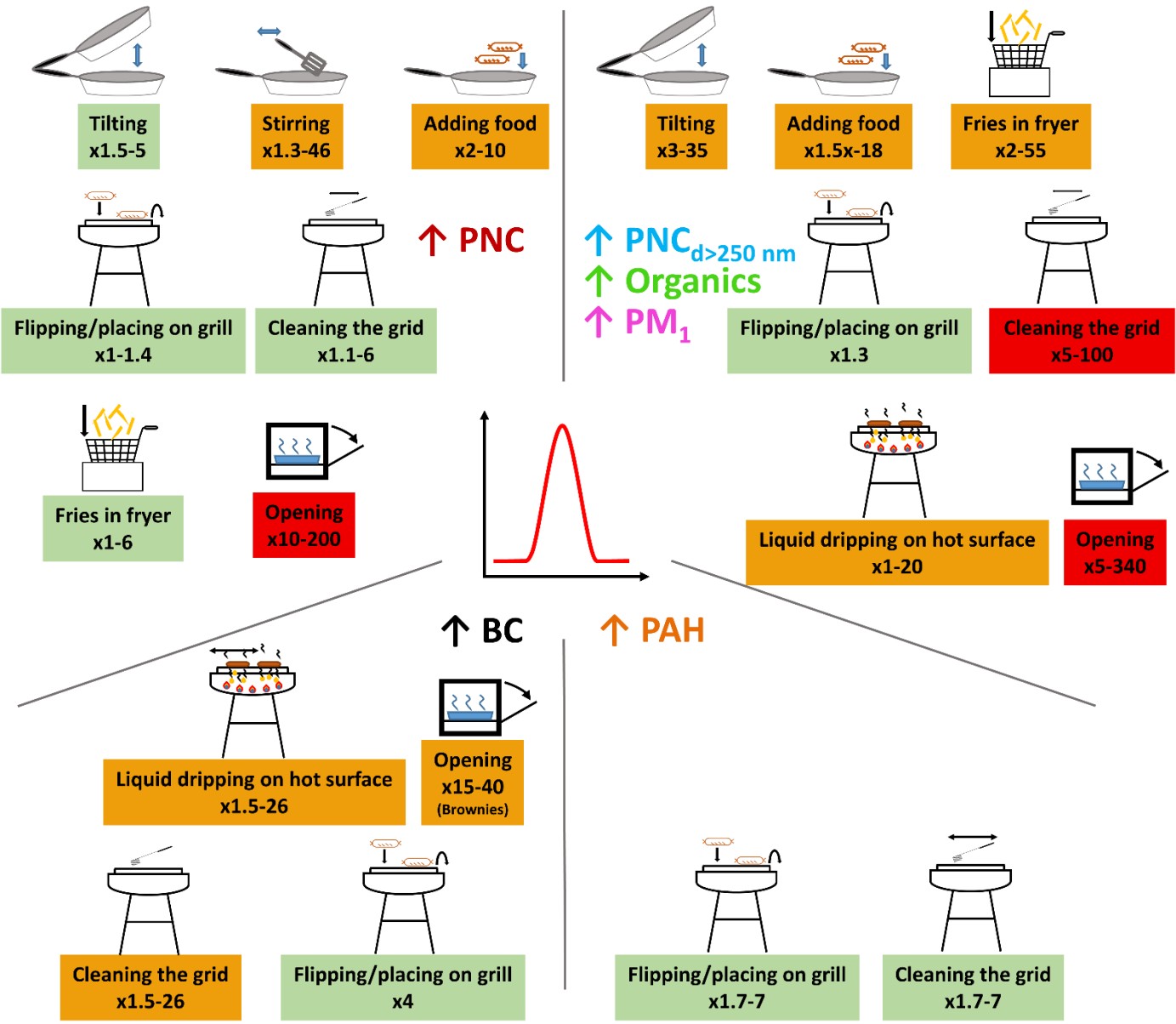

**Figure 8: Schematic diagram of the short-term concentration increases for the different variables due to various activities during the cooking of the dishes. PM$_1$ is shown as representative of PM. The range of factors by which concentrations typically increase is shown as numbers and color-coded, green for small, orange for medium, and red for high concentration increases.**

### 3.3 Influence of cooking method and cooking activities on the particle size distribution

The averaged particle number and volume size distributions of the emitted aerosols were similar for dishes with the same cooking method in terms of particle mode position and intensity. An overview of the mode diameters for the aerosols emitted during the cooking of different dishes, grouped by cooking method or dish type, is shown in Table 5. The average standard deviation of the mode diameters from the three replicates was 5 nm for the particle number size distribution and 25 nm for the particle volume size distribution. Therefore, several of the observed differences between the distributions for the different cooking methods were statistically significant.

The particle number distribution for most dishes was dominated by Aitken mode particles. The mode diameters ($d_{p,N}$) varied between 20 and 50 nm depending on the cooking method (Fig. S6). During the warm-up phase of the grilling experiments, the size distribution was broader and plateau-like, presumably due to a combination of different particle generation processes, such as combustion of grid residues and incomplete combustion of charcoal, but also dominated by Aitken mode particles (10 – 30 nm).

The average volume size distributions showed more variability for the different cooking methods (Fig. S7). The distributions were mostly bimodal with an Aitken or accumulation mode and a coarse mode. For baking and grilling with gas, the mode diameter of the fine particles was in the Aitken mode range ($d_{p,V} = 50 - 70$ nm) while for frying and grilling with charcoal the distribution was dominated by accumulation mode particles (200 – 300 nm). The coarse mode diameter was in the range of 2 – 3 µm.

**Table 5: Range of mode diameters from the averaged particle number and volume size distributions for particles emitted from cooking different dishes, sorted by mode diameter (dN/dlog$d_p$).**

| Cooking method/ dish type | Dishes | Mode diameter dN/dlog$d_p$ ($d_{p,N}$) | Mode diameter dV/dlog$d_p$ ($d_{p,V}$) |
|---|---|---|---|
| Grill warm up (gas, charcoal) | | 20 — 30 nm | Gas: 50 — 60 nm, 2.5 — 3 µm<br>Charcoal: 300 nm, 720 nm, 2.2 µm |
| Deep-frying in pot | French fries, Bavarian doughnut | 20 — 30 nm | 275 — 280 nm, 2 µm |
| Stir-frying with sauce | Spaghetti Bolognese, stir-fried vegetables, Indian curry | 20 — 35 nm | 205 — 220 nm, 2 — 3 µm |
| Grilling with gas | Vegetable skewers, steak | 30 — 35 nm | 60 — 70 nm, 2 — 5 µm |
| Baking | Baked potatoes, pizza, brownies | 30 — 35 nm | 45 — 70 nm, 2 — 3 µm |
| Stir-frying | Fried potatoes, bratwurst, schnitzel, fish | 40 — 50 nm | 205 — 220 nm, 2 — 3 µm |
| Deep-frying in deep fryer | French fries | 50 nm | 205 nm, 2 — 3 µm |
| Grilling with charcoal | Steak | 50 nm | 205 nm, 600 nm, 2.2 µm |
| Boiling | Boiled potatoes, rice, noodles | No clear result due to small concentrations | 300 — 465 nm |

Presumably, the observed mode diameter of the emitted fine (i.e., submicron) aerosol is most affected by the temperature of the prepared food and cookware. Higher temperatures allow more oil and other substances to vaporize, resulting in greater particle
growth and consequently larger particles. For example, particles from stir-fried dishes were larger ($d_{p,N} = 40 – 50$ nm) than those from stir-fried dishes with sauce (20 – 35 nm) because the addition of the sauce cooled the food and pan and the sauce effectively covered the hottest part of the system, the bottom of the pan. In addition, the amount of material available for vaporization affects the particle growth. For example, frying has more oil available to vaporize compared to baking, where it is limited to the dough components, resulting in larger particles. Charcoal grilling produces larger particles than gas grilling because the incomplete
combustion of charcoal produces smoke and the higher temperature allows additional material to vaporize, including from the charcoal itself.

The coarse mode particles are generated by mechanical processes, presumably by the bursting of oil bubbles. When grilling with charcoal, the combustion of the charcoal also results in the emission of coarse particles. The particles emitted from the boiled dishes are probably initially coarse particles from the bursting of water bubbles with droplets containing dissolved salt and other
food components, which shrink to accumulation mode particles due to the low relative humidity.

Consistent with our measurements, similar dependencies of mode diameter on temperature and available amount of vaporizable material have been observed in previous studies. With increasing cooking temperatures Amouei Torkmahalleh et al. (2012), Buonanno et al. (2009), and Zhang et al. (2010) measured particle size distributions with larger mode diameters. Furthermore,

Buonanno et al. (2009) observed large number mode diameters ($d_{p,N}$ = 40 – 50 nm) for emissions from grilling (without oil on an electric or gas grill) of fatty foods, such as cheese, bacon, and sausage, compared to those from cooking vegetables ($d_{p,N}$ = 30 nm) showing that the availability of easily vaporizable substances, in this case fat or its decomposition products, leads to larger particles. In addition to the cooking method, which is mainly characterized by the cooking temperature and the availability of water, oil, or fat, individual activities during cooking also influence the particle size distribution of the emitted aerosol. Such influences are illustrated in Fig. 9 using the example of frying French fries in a deep fryer, showing the number size distributions (15 - 30 s time periods, averaged over all replicates) of emissions during different activities or cooking phases. The corresponding PM$_1$ mass concentrations for the same time periods are also shown; colored arrows indicate the temporal changes.

Initially, the particle number concentration, size, and mass concentration increase as the frozen French fries are placed in the basket above the oil and then submerged in the oil (light red arrow). When the fries are placed in the basket, the oil begins to bubble as small pieces of the fries and ice crystals fall into the hot oil and the water immediately vaporizes. The bubbling increases when the French fries are submerged in the oil as more water vaporizes quickly. The bubbles increase the surface area of the oil, which increases the vaporization of the oil and the formation and growth of particles. As a result of the frozen French fries in the oil, the oil cools and less oil vaporizes, resulting in a decrease in particle number concentration, size, and mass concentration (red arrow). As the oil slowly heats up again towards the end of the cooking process, all variables increase again due to increased oil vaporization (dark red arrow).

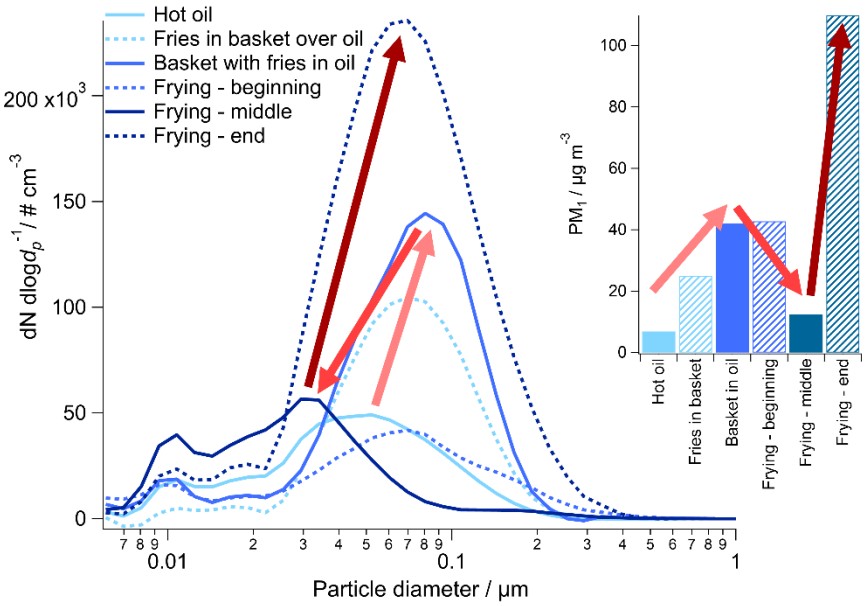

**Figure 9: Average number size distribution and PM$_1$ mass concentration for six different cooking activities/periods during the cooking of French fries in a deep fryer. The arrows indicate the temporal trends.**

The example presented illustrates the main parameters that influence particle emissions: 1. the temperature of the prepared food and cookware, 2. the oil surface, and 3. the available amount of vaporizable material, as also observed for the particle number concentration and mass concentration for different variables (see Sect. 3.2). Similar dependencies were also observed during the cooking of other dishes (Table 6). In general, the mode diameter increased during the cooking time, as observed for example during heating of the oven and charcoal grilling. Presumably, the increase in temperature of the food and the cookware led to stronger vaporization of oil and other substances. Also, various activities during food cooking resulted in transient changes in the size of the emitted particles, analogous to the changes in emission intensity, as discussed in Sect. 3.2. In addition, when the grid of the grill was cleaned with a brush, the particle size increased, presumably because leftovers fell from the grid onto the charcoal and

burned or vaporized. A similar process was observed when steaks were cut on the grill and the meat juices vaporized from the hot grid or charcoal, also resulting in larger particles.

**Table 6: Overview of particle mode diameter changes due to individual activities.**

| Process/activity | Mode diameter dN/dlog$d_p$ | Reason |
|---|---|---|
| Charcoal grilling | 35 nm → 170 nm | Temperature increase over time |
| Stir-frying | 30 nm → 60 nm | Temperature increase over time |
| Heating of oven | 17 nm → 40 nm | Temperature increase over time |
| Cleaning the grid of the grill | Increase by 5 – 10 nm | Food residues from the grid vaporized on hot surface |
| Cutting steaks on grill | Increase by 5 – 10 nm | Meat juices vaporized from hot surface |

### 3.4 Quantification of cooking emissions: Emission factors

In order to quantitatively estimate the emissions from cooking activities and their impact on air quality based on the mass of food prepared, emission factors (amount of emitted substance per kg of food prepared) were calculated for all dishes from this study and for all relevant variables (Table S6). The emission factors for PN (particle number, as measured by the CPC) and $PM_1$ are
shown in Fig. 10 as examples for all dishes, grouped according to the respective cooking method. For other mass-based variables, such as organics, the general trends are similar to those for $PM_1$ and are described below.

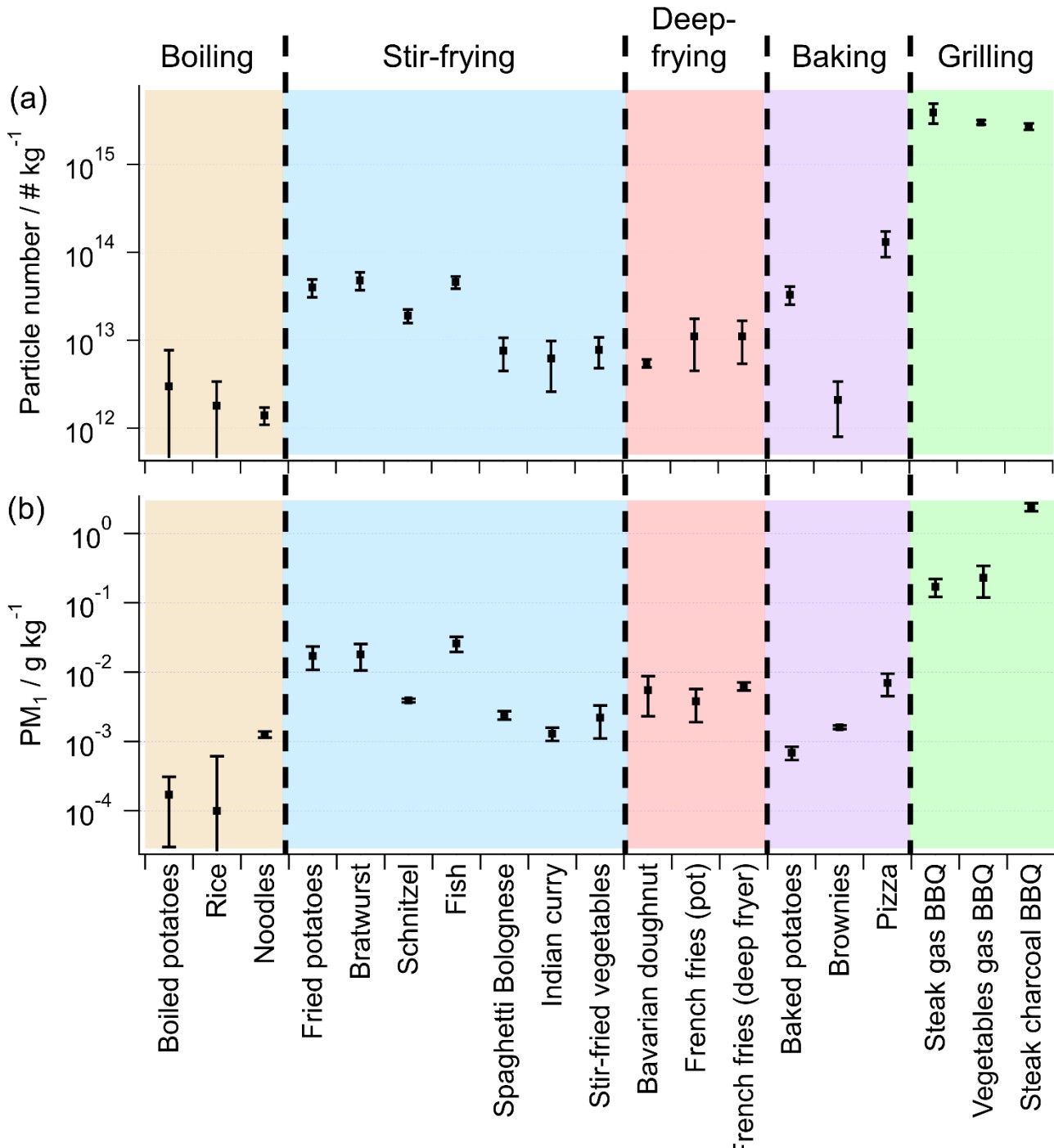

**Figure 10: Emission factors for (a) PN and (b) PM₁ for all dishes, with the standard deviation of the three replicates as error bars. The values are grouped by cooking method, highlighted in different colors.**

For dishes with the same cooking method, the emission factors are similar and differ by at most one order of magnitude. The highest PN emission factors were observed for the grilling experiments with values up to $4 \cdot 10^{15}$ kg$^{-1}$, while the emission factors for the oil-based or fat-containing dishes, including the cooking methods stir-frying, deep-frying, and baking, are substantially smaller, ranging from $2.1 \cdot 10^{12} - 1.3 \cdot 10^{14}$ kg$^{-1}$. The smallest emission factors were observed for boiled dishes with values up to $3 \cdot 10^{12}$ kg$^{-1}$.

A similar trend was observed for $PM_1$ with the highest emission factors for the grilling experiments ($0.2 - 2.4$ g $kg^{-1}$) and one to two orders of magnitude lower emission factors for stir-fried, deep-fried and baked dishes ($7 \cdot 10^{-4} - 0.026$ g $kg^{-1}$). Again, the smallest emission factors were found for boiled dishes ($1 \cdot 10^{-4} - 1.3 \cdot 10^{-3}$ g $kg^{-1}$).

The $PN_{d>250 \, nm}$ (number of particles measured by the OPC, i.e. with $d_p$ >250 nm) emission factors range from $5 \cdot 10^7 - 2 \cdot 10^{10}$ $kg^{-1}$ for boiled and baked dishes, from more than $2 \cdot 10^{10} - 9 \cdot 10^{11}$ $kg^{-1}$ for stir-fried, deep-fried, and gas-grilled dishes, and up to

$2 \cdot 10^{13}$ $kg^{-1}$ for the charcoal-grilled dish. BC and PAH emissions were observed only for dishes where the cooking temperatures were sufficiently high for their formation, e.g. the grilling and stir-frying experiments ($18 - 28{,}000$ µg $kg^{-1}$ and $3 - 208$ µg $kg^{-1}$, respectively). Sulfate was observed only for dishes with onions and for grilled dishes ($6 - 354$ µg $kg^{-1}$). Emission factors for all variables are listed in Table S6.

In general, the trends in the observed emission factors for the different cooking methods were similar for the different measured

variables. For mass-based or related variables ($PM_1$, organics, PAH, BC, and $PN_{d>250 \, nm}$), the emission factors from the charcoal grilling experiment are typically one order of magnitude higher than those from the gas grilling experiments. The incomplete combustion of the charcoal results in the additional emission of smoke containing larger particles and a higher total emitted mass. The combustion of the charcoal during the heating of the grill already contributes $34 - 52\%$ of the total emissions for the whole cooking experiment, depending on the variable (PN, $NO_x$, organics: $34 - 40$ %; PAH, $PM_{1/2.5/10}$, $PN_{d>250 \, nm}$: $40 - 50$ %; BC: $52$ %).

The emissions from grilling are one to two orders of magnitude higher than those from other cooking methods, presumably due to the burning of food residues on the grid and the higher temperatures, which lead to more vaporization of substances and thus to increased particle formation and growth due to re-condensation.

The emission factors for stir-fried, deep-fried, and baked dishes were similar, since in these cases the emissions are mainly due to the vaporization and recondensation of oil and other substances, as well as mechanical processes such as the vaporization of water,

which leads to oil bubbling and splashing. The lowest emissions were observed for boiled dishes, which was the only cooking method used that did not involve oil or fatty foods. In this cooking method, the only source of particles is the bursting of bubbles, which results in droplets containing dissolved salt or other components.

Oil-based cooking (e.g. deep-frying and stir-frying) causing higher particle number concentrations compared to water-based cooking (boiling and steaming) has also been observed by See and Balasubramanian (2006), Wu et al. (2012), and Zhang et al.

(2010). Similar observations were made for emitted particle mass (Alves et al., 2014; See and Balasubramanian, 2006) and PAH emissions (Chen et al., 2007; Zhao et al., 2019).

For comparison with the results of previous studies, PN and $PM_{2.5}$ emission rates (Table 7) were calculated for 1 kg of cooked food and 60 min of cooking time (assuming that food preparation takes one hour) for different cooking methods. The emission rates determined from our experiments were mostly comparable to those obtained in previous studies (He et al., 2004; Liao et al., 2006)

or agreed with them within one order of magnitude (Lee et al., 2001; Nasir and Colbeck, 2013). In contrast, Buonanno et al. (2009) reported emission rates up to two orders of magnitude higher for PN and Olson and Burke (2006) for $PM_{2.5}$.

**Table 7: PN and PM$_{2.5}$ emission rates for 1 kg of cooked food per 60 min of cooking time, for different cooking methods. Comparison of our results with those of previous studies.**

| | PN / kg$^{-1}$ h$^{-1}$ | PM$_{2.5}$ / mg kg$^{-1}$ h$^{-1}$ |
|---|---|---|
| **Stir-frying** | | |
| This work | $5.2 \cdot 10^{13}$ | 23 |
| Buonanno et al. (2011) | $4.5 \cdot 10^{15} - 5.4 \cdot 10^{15}$ | |
| Nasir and Colbeck (2013) | $8 \cdot 10^{12}$ | 78 |
| He et al. (2004) | $1.5 \cdot 10^{13}$ | |
| **Baking** | | |
| This work | $8.6 \cdot 10^{13}$ | 5 |
| Nasir and Colbeck (2013) | $2.6 \cdot 10^{13}$ | 45 |
| He et al. (2004) | $1.2 \cdot 10^{13}$ | |
| Olson and Burke (2006) | | 600 |
| **Grilling** | | |
| This work | | $280 - 2700$ |
| Olson and Burke (2006) | | 10380 |
| **Deep-frying** | | |
| This work | | 10 |
| Liao et al. (2006) | | $3.2 - 8$ |
| Lee et al. (2001) | | 70 |
| Olson and Burke (2006) | | 3600 |


In the case of the study by Buonanno et al. (2011), these differences may be due to different measurement conditions, as the emissions in that study were measured in a closed kitchen with mechanical ventilation, at a distance of 2 m from the stove and not by capturing all emissions as in our study. In the case of the study by Olson and Burke (2006), who performed measurements with body-worn instruments to assess personal exposure, the massively higher emission rates they found compared to our and previous

studies were presumably due to a combination of reasons, such as the influence of the high relative humidity on the measured particle mass, their assumptions about dilution of the emissions, and the use of peak concentrations for their calculation rather than averages over the entire experiment.

Overall, the comparison of emission rate measurements shows that the emission rates obtained depend not only on the cooking conditions themselves, but also on the measurement (dilution) conditions and the method used to calculate the emission factors or

rates. This makes it difficult to compare different studies.

To obtain an idea of the relevance of emissions from cooking activities in relation to those from other emission sources, the emissions from the various cooking methods were compared with emissions from traffic, biomass burning, burning of candles, and smoking. For this purpose, the emissions from these sources were calculated for activities over a period of one hour, i.e. for the one-time cooking of a meal ("cooking"), for driving a car over a distance of 100 km ("driving a car"), for smoking two cigarettes

("smoking"), and for wood burning-based heating a room of 50 m$^2$ ("wood home heating") or for burning a candle ("candle burning") for one hour. The emission factors for the various activities were taken from the literature and are summarized in Table S7. As these activities are partially arbitrarily chosen, this comparison only serves as a rough classification of cooking emissions compared to those of other emission sources.

The calculated emissions for the dishes with the same cooking method were averaged for four variables: PM$_1$, PN, BC, and PAH

(Fig. 11), and their standard deviation is used as the uncertainty. For the emissions from other sources, the ranges of emissions

calculated from the emission factors found in the literature are presented as bars to reflect the variability of the emission levels.

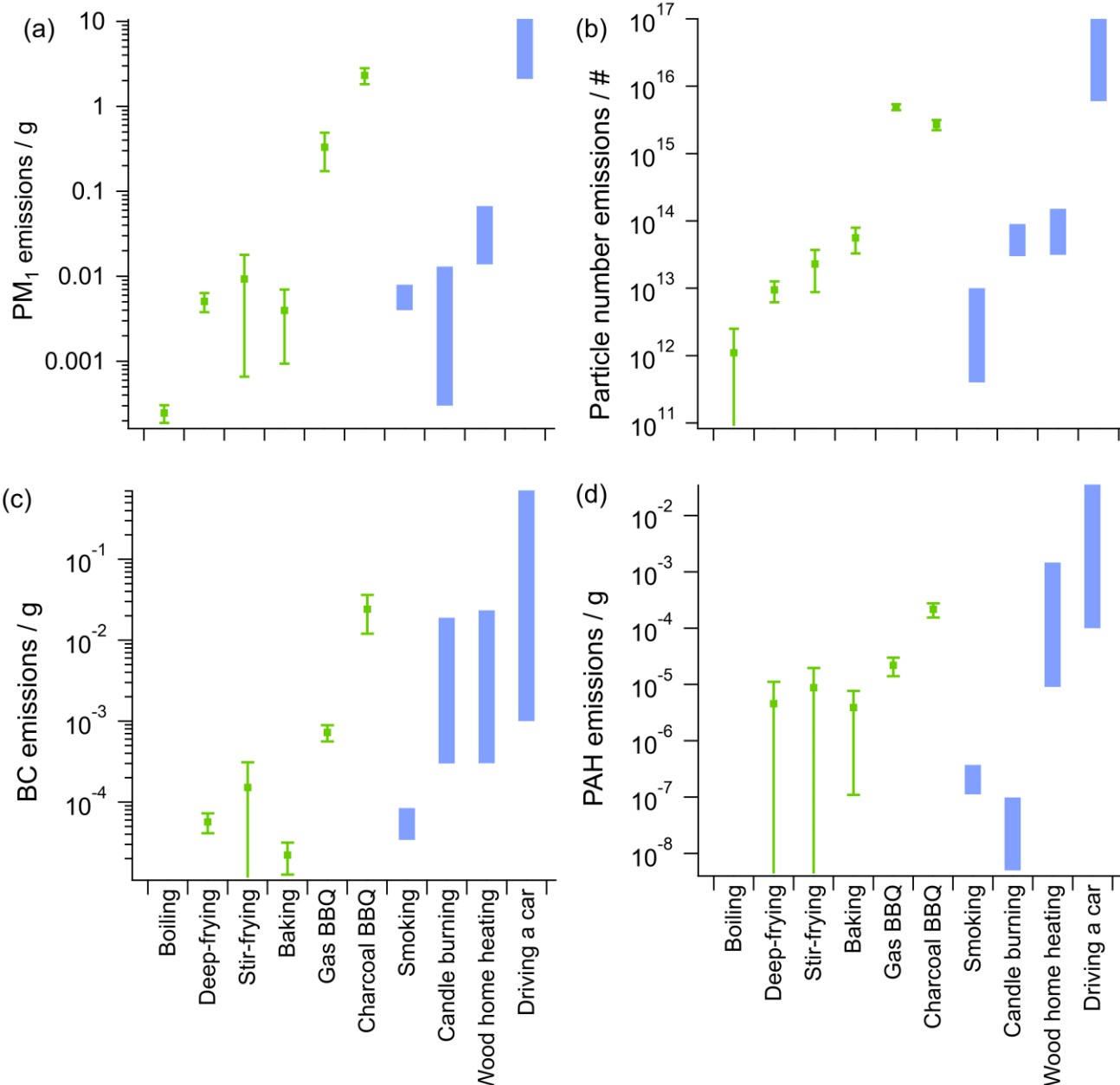

**Figure 11: Total emissions per unit activity of (a) PM$_1$ mass, (b) particle number, (c) black carbon mass, and (d) PAH mass for cooking one dish, averaged for the different cooking methods with the standard deviation as error bars, and comparison with emissions from**
**various other activities during one hour, shown as bars indicating the variability found in the literature (Table S7).**

For the mass-based variables (PM$_1$, BC, PAH), the highest cooking emissions, which were from charcoal grilling, are in the same

range as those observed from car driving, indicating the potential for a substantial local impact of grilling on air quality. This

assumption is supported by a study by Kaltsonoudis et al. (2017), which shows that during a Greek holiday, when meat is

traditionally grilled all over the city, the contribution of COA reached up to 85% of the measured organic aerosol.

Stir-frying, deep-frying, and baking, all oil-based cooking methods, have emissions of similar order of magnitude, typically at the

lower end of emissions from wood burning-based room heating and at the upper end of emissions from candle burning and cigarette

smoking. This finding is consistent with observations from ambient measurements, which show that COA can easily account for

similar proportions of total organics as traffic- and wood burning-related organic aerosols, particularly in urban environments (e.g.,

Mohr et al., 2012; Struckmeier et al., 2016). In indoor environments, cooking is one of the major emission sources leading to high

fine particulate matter emissions in terms of number and mass, even exceeding emissions from light smoking (Abdullahi et al.,

2013; Zhou et al., 2016; He et al., 2004).

Boiling, on the other hand, results in much lower emissions, at the lower end or even below those of smoking and candle burning.

Thus, unlike oil-based cooking methods, boiling will typically be not a major contributor to the total ambient aerosol load, which

is consistent with the conclusion that ambient COA consists mainly of externally mixed (Freutel et al., 2013) oil or fatty acids

containing droplets (Allan et al., 2010).

## 3.5  Ambient measurements at two Christmas markets

At both Christmas markets, substantial increases in aerosol concentrations were measured during the opening hours compared to

the background (i.e., the hours when the markets were closed) for the same six species that were relevant in the laboratory

measurements: PNC and $PNC_{d>250\ nm}$, PM, BC, PAH, and organics mass concentrations (Fig. S8 and Fig. S9). In addition, $CO_2$ and

particulate chloride concentrations increased, particularly at the market in Ingelheim, both presumably due to wood burning at the

market (Fachinger et al., 2018; Levin et al., 2010; Williams et al., 2012). A summary of the measured concentrations (shown in

box plots) for time periods within and outside of opening hours is shown in Fig. 12, which illustrates the increase in concentrations

due to the Christmas market emissions.

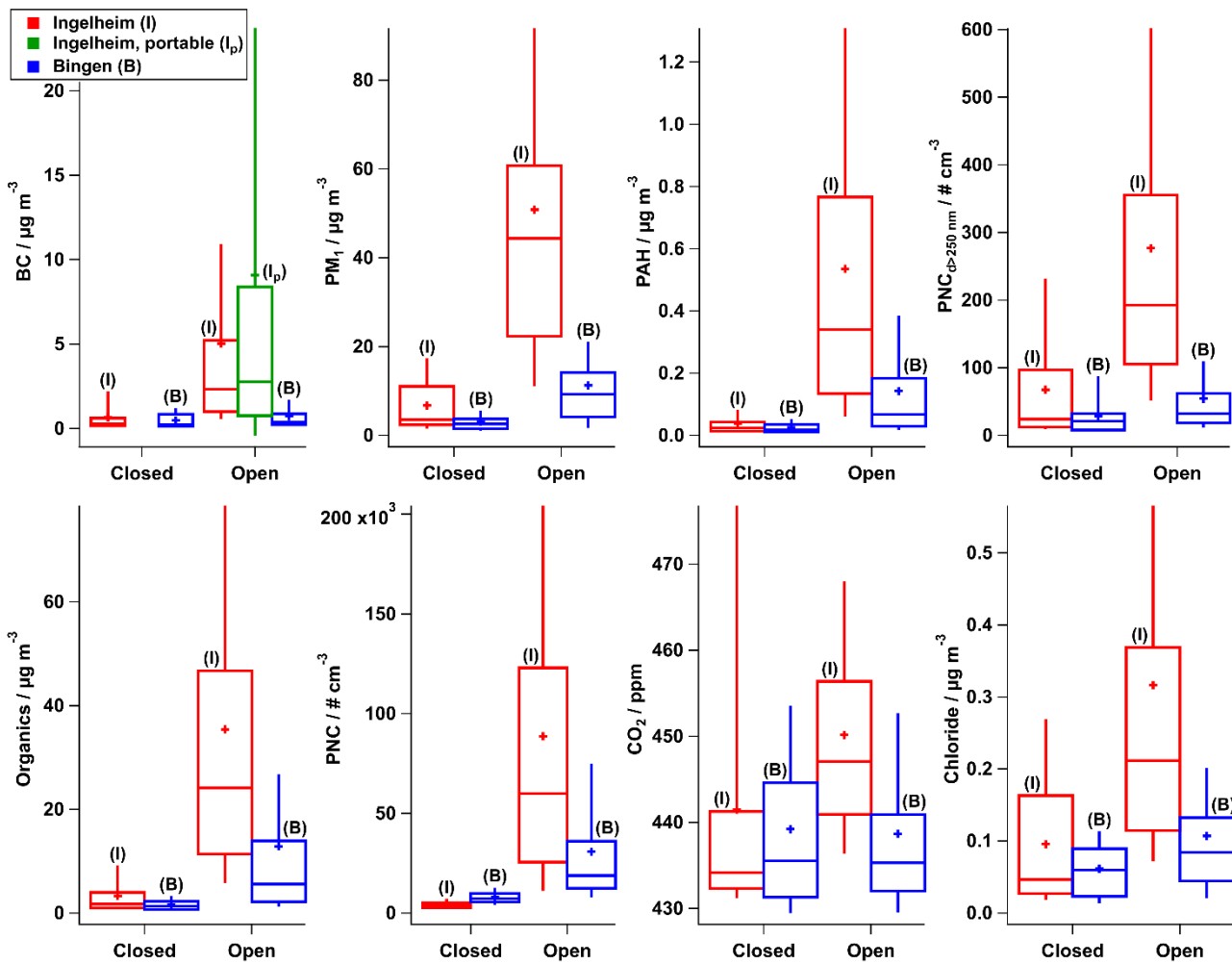


**Figure 12: Pollutant concentrations measured during (open) and outside (closed) opening hours of the Christmas markets in Ingelheim (red and green) and Bingen (blue). For each variable, the average concentration is shown as cross, the 25th and 75th percentiles as a box, the median as a horizontal bar, and the 10th and 90th percentiles as whiskers.**

In Ingelheim, the median PNC and the mass concentrations of organics, $PM_1$ and PAH were more than one order of magnitude

higher during the opening hours than during the background period. The median $PNC_{d>250\,nm}$, BC, and particulate chloride mass concentrations were increased by a factor of $4 - 8$. The median $CO_2$ volume mixing ratio was 13 ppm higher. In Bingen, the median concentration enhancements due to Christmas market emissions were smaller: for organics, $PM_1$, and PAH mass concentrations by a factor of $3.5 - 4.5$, for the other variables by a factor of $1.5 - 2.5$, except for $CO_2$, which did not show an increase during the opening hours.

The different concentration levels between the two locations during opening hours are presumably due to two reasons. First, the monitoring site in Ingelheim was very close (a few meters) to the food stands, while the distance to the nearest food stand in Bingen was about 25 m. Second, the Christmas market in Ingelheim was larger, with more visitors and more densely packed food stands. In general, the measurements show that emissions from a Christmas market can lead to substantial increases in pollutant concentrations at the local level.

In Ingelheim, BC mass concentrations were additionally measured with a portable aethalometer (Fig. 12, green box plot for BC) while repeatedly walking across the Christmas market during opening hours in order to estimate the personal exposure of market visitors. The median of these mobile measurements across the market was similar to the median of the stationary measurements

directly downwind the market. This indicates that the concentrations measured at a single location on the downwind edge are representative of the market as a whole. At the same time, the average concentration measured with the portable instrument (9.1 µg m$^{-3}$) was almost twice as high as the average of the stationary measurements (5.0 µg m$^{-3}$). Thus, visitors to the market may be exposed to much higher transient BC concentrations, presumably when walking near fireplaces or other strong sources, thereby increasing their personal exposure.

### 3.5.1 PMF analysis of the AMS organics data

For detailed information on the contribution of different aerosol types, the AMS organics mass spectra were analyzed using positive matrix factorization (PMF), separately for both Christmas markets. For both markets, BBOA, COA, and OOA (usually associated with aged background aerosol) were identified as aerosol types from the most reasonable PMF solution (Figs. S10 and S11). The challenge in this analysis was that two emission sources, cooking and biomass burning, were close to each other with similar activity times, while a requirement for the PMF algorithm to separate different aerosol types is a characteristic temporal variation that is different for each aerosol type. This resulted in an incomplete separation of the OOA factor for the measurements in Ingelheim with considerable OOA concentration increases during the opening hours of the market, while for this background related aerosol type rather constant concentrations independent of the opening times are expected (as seen in Bingen).

The mass spectra of COA, BBOA, and OOA are similar for both sites and show the typical markers for each aerosol type. In the mass spectra of OOA the most intense signal is at $m/z$ 44 (CO$_2^+$) due to thermal decomposition of oxidized organic compounds (Ng et al., 2010). BBOA could be identified by the elevated signal intensities at $m/z$ 60 and 73, whose ratio of 2.6 at both markets points to levoglucosan (see Sect. 3.1.2), resulting from the pyrolysis of cellulose (Schneider et al., 2006). In the COA mass spectra, the highest signal intensities are at $m/z$ 41 and 55, and the signal ratio of $m/z$ 55 and 57 is 2.6, which is consistent with the results of previous studies (Mohr et al., 2012; Sun et al., 2011; Xu et al., 2020) and our laboratory studies (Sect. 3.1.2). Correlation with corresponding reference mass spectra (averaged from the available mass spectra from the AMS database, see Table S3) supported the assignment of the identified factors, with correlation coefficients of 0.93 and 0.97 for COA, 0.98 and 0.95 for OOA, and 0.83 and 0.77 for BBOA for Ingelheim and Bingen, respectively.

COA and BBOA concentrations increased substantially during the opening hours, while OOA concentrations remained almost constant (OOA for Ingelheim not considered here due to incomplete separation). The average concentrations of COA (CE = 1; RIE = 2.27; see Sect. 3.5.2) were 3.5/0.14 µg m$^{-3}$ and 2.5/0.05 µg m$^{-3}$ and of BBOA (CE = 0.5; RIE = 1.4) were 17.1/0.54 µg m$^{-3}$ and 2.4/0.21 µg m$^{-3}$ during/outside opening hours for Ingelheim and Bingen, respectively. In Bingen, the OOA concentration (CE = 0.5; RIE = 1.4) was mostly below 2 µg m$^{-3}$ during the whole measurement period, suggesting that this PMF factor can be attributed to the background aerosol. The observed step changes in OOA concentration (Fig. S10) were due to changes in wind direction. The fraction of OOA at both Christmas markets during the opening hours was similar at 15 % and 17 %, while the fraction of BBOA was 71 % and 40 % and the fraction of COA was 14 % and 43 % for Ingelheim and Bingen, respectively. The higher proportion of BBOA in Ingelheim may be due to a second wood-fired barrel 25 m away from MoLa on two afternoons and a flame-grilled salmon stand with an open wood fire within the circle of food stands where MoLa was located.

### 3.5.2 Validation of laboratory measurements using the Christmas market data

To assess whether the results of the laboratory experiments are also applicable to ambient measurements, we used the Christmas market data to verify several aspects of our results. Due to the higher fraction of COA measured during the Christmas market opening hours in Bingen (43 %) compared to Ingelheim (14 %) the analysis was performed only with the data set collected in Bingen.

The dishes prepared at the Christmas market which were also investigated in the laboratory are fried bratwurst, fried French fries (in a deep fryer), and steaks grilled on a gas grill. A linear correlation of the average COA mass spectrum from the PMF analysis of the Christmas market data with the mass spectra of the above three dishes showed a very high similarity between the spectra (Pearson's $r = 0.99$), as did the correlation with those of rapeseed oil ($r = 0.98$) and oleic acid ($r = 0.93$). The ratio of $f_{67}/f_{69}$ for this COA mass spectrum was 1.4, similar to the ratios of previously measured ambient COA ($1.2 \pm 0.1$) and laboratory measurements ($1.1 – 1.6$), supporting our proposal of $f_{67}/f_{69}$ as an additional COA marker (see Sect. 3.1.2).

To verify whether the densities for the organic fraction derived from the cooking emission experiments can be applied to ambient measurements, the densities for the three Christmas market-related dishes as well as for the COA PMF factor from the market measurements were calculated based on the formula of Kuwata et al. (2012). The density of COA at 0.94 g cm$^{-3}$ is consistent with the densities for the three dishes ($0.94 – 0.98$ g cm$^{-3}$, Table S5). This finding, together with the high mass spectral similarity discussed above, suggests that the observed ambient COA is composed to a substantial amount of vaporized and recondensed oil or decomposed fats.

To validate whether the RIE$_{COA}$ values determined from laboratory measurements are applicable to ambient measurements of cooking related aerosols, PM$_1$ (from FMPS and OPC measurements, Sect. S1) was compared with PM$_1$ calculated from BC and AMS species (PM$_{1,AMS+BC}$) for two different sets of RIE$_{COA}$ and CE$_{COA}$ values: i) the standard AMS values, i.e. RIE$_{COA}$ = 1.4 and CE$_{COA}$ = 0.5 (Fig. 13a); and ii) average values derived from the laboratory measurements of the three Christmas market-related dishes (RIE$_{COA}$ = 2.27 and CE$_{COA}$ = 1; Fig. 13b). In addition, the proportions of the different aerosol species in the Christmas market PM$_1$ emissions (after background subtraction) are shown as pie charts in Fig. 13, calculated by applying the corresponding RIE and CE values to the COA. In both cases, default RIE and CE values were used for the other AMS species including BBOA and OOA (i.e., assuming externally mixed COA; Freutel et al., 2013). As illustrated in Fig. 13, the correlations between the two types of PM$_1$ values are characterized by a considerable amount of scatter, particularly in the lower PM$_1$ concentration range. This is likely due to the fact that several sources for cooking-related PM$_1$ as well as for other types of organic aerosol are in close proximity to the measurement location, resulting in substantial variability in the data from the instruments used to determine PM$_1$. This is also reflected in the poor correlation coefficients for both approaches to calculate PM$_1$ from AMS and BC data ($r^2 = 0.56$ and 0.58 with the default and laboratory values for RIE$_{COA}$ and CE$_{COA}$, respectively). Figure 13a illustrates that PM$_{1,AMS+BC}$ appears to be overestimated for higher PM$_1$ concentrations when the default values are employed. In contrast, when RIE$_{COA}$ and CE$_{COA}$ are derived from the laboratory results, PM$_1$ values scatter more around the one-to-one line (Figure 13b), suggesting improved mass closure. ODR fitting of the two pairs of PM$_1$ data with the intercept forced through the origin yields PM$_{1,AMS+BC}$ = 1.71 * PM$_1$ and PM$_{1,AMS+BC}$ = 0.68 * PM$_1$, respectively, for the default and the laboratory values. These results indicate a slight improvement in the agreement between the two sets of data when the laboratory RIE$_{COA}$ and CE$_{COA}$ values were employed. The pie charts show the effect of the different RIE and CE values on the calculated fraction of COA of the emitted Christmas market PM$_1$. Using the default values, the COA fraction would be 26% higher compared to using the laboratory values, showing the importance of choosing correct RIE and CE values for COA.

In general, the result of this comparison is consistent with previous ambient measurements of cooking emissions, which also suggest a higher RIE$_{COA}$ value than the standard RIE$_{Org}$ of 1.4 (Katz et al., 2021; Reyes-Villegas et al., 2018).

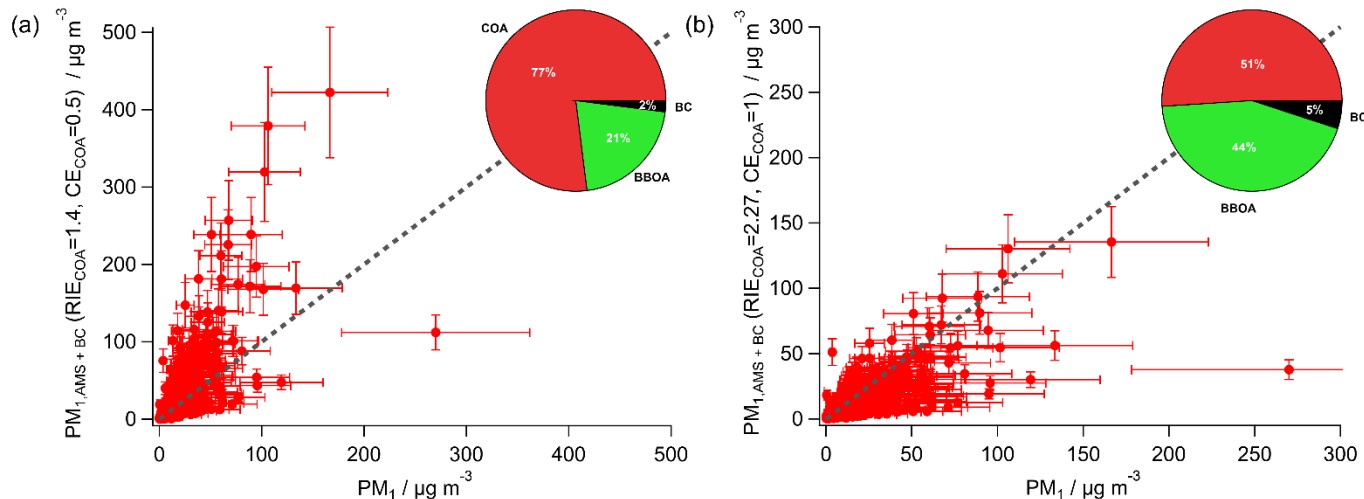

**Figure 13: Comparison of measured PM$_{1,\text{AMS+BC}}$ with PM$_1$ using (a) RIE$_{\text{COA}}$ = 1.4, CE$_{\text{COA}}$ = 0.5 and (b) RIE$_{\text{COA}}$ = 2.27, CE$_{\text{COA}}$ = 1 for the COA fraction. The 1:1 line serves as a guide for the eye. The pie charts show the calculated PM$_1$ composition of the Christmas market emissions (i.e., only for opening hours, after background subtraction).**

Based on the results of the laboratory experiments as well as those of previous studies, no strong contribution of BC from cooking emissions was expected (Zhang et al., 2010; Zhao et al., 2007), and the observed BC was assumed to originate mainly from biomass burning. In fact, the ratio of BBOA (RIE = 1.4 and CE = 0.5) to BC mass concentrations was on average 3.3 during the Christmas market opening hours, which is well within the range of 1.7 – 33 observed for open biomass burning (Reid et al., 2005) and close to the ratios of 4.0 and 3.16 measured by Crippa et al. (2013) and Elser et al. (2016) for mainly residential heating in urban environments.

The applicability of the laboratory emission factors (see Sect. 3.4) to ambient measurements was verified by testing whether they could reproduce the concentrations measured during the Christmas market in a simple model. For this purpose, the emission factors determined in the laboratory for the variables PN, PM$_1$, and organics were used for the dishes which were prepared at the Christmas market (fried bratwurst, French fries fried in a deep-fryer, and steaks from a gas grill). Here, we assume that the emission factors obtained in the laboratory for the marinated steak are not strongly different from those for the non-marinated steak, which was used for cooking on the Christmas market. The emission factors for gas grilling were used instead of those for charcoal grilling, because PMF is likely to allocate part of the charcoal grilling to the biomass burning factor, leading to an underestimation of the respective COA emissions.

The emissions per hour ($EM$) needed to generate the measured concentrations were calculated using the average concentration during the opening hours ($\overline{c_{CM}}$) minus the average background concentration ($\overline{c_{BG}}$) and the volumetric flow rate $Q_{CM}$ with which the emissions were diluted (Eq. (3)). The volumetric flow rate $Q_{CM}$ was estimated based on the average wind speed (1.15 m s$^{-1}$, mostly from the west), the height of the houses surrounding the square (8 m), to which we assumed the emissions would be diluted, and the width of the street running from west to east, which transports most of the air mass, resulting in $Q_{CM}$ = 5 · 10$^5$ m$^3$ h$^{-1}$ (138 m$^3$ s$^{-1}$). Using the emission factors $EF$ from the laboratory experiments, we calculated the amount of food ($m$) that would need to be cooked per hour to generate the calculated emissions per hour (Eq. (4)).

$$EM = (\overline{c_{CM}} - \overline{c_{BG}}) \cdot Q_{CM} \tag{3}$$

$$m = \frac{EM}{EF} \tag{4}$$

Finally, assuming that a bratwurst has a mass of 150 g, a schnitzel has a mass of 180 g, and a unit of French fries has a mass of 250 g, the calculated masses were converted into food units to make the results more tangible. Since the emission factors were determined from cooking activities, only the COA-related fraction of the measured Christmas market emissions was considered for the mass-based variables $PM_1$ and organic mass concentration. The COA concentration was calculated using $RIE_{COA} = 2.27$ and $CE_{COA} = 1$ and considering only the Christmas market emissions (background subtracted). For $PM_1$, the fraction related to COA is 51% (Fig. 13b), and for the total measured AMS organics it is 54%. Since it is not possible to determine the COA related fraction for PN based on the $PM_1$ results, we assumed that the COA related fraction for PN would be somewhere between 20% and 80% and performed the calculations for these two extreme scenarios.

Figure 14 shows, for the three selected variables, the number of food units that would need to be cooked per hour of each dish to account for the observed emissions. For the mass-based variables, the calculated numbers of steaks were 67 – 94 per hour, and for bratwurst and French fries, the numbers were at least an order of magnitude higher at 770 – 2150 units per hour. For PN, the calculated numbers of food units for the chosen COA fraction range of 20% to 80% were smaller than those for the mass-based variables for steaks at 3 – 13 units per hour and similar to those for the mass-based variables at 310 - 1250 units of bratwurst and 800 – 3200 units of French fries. These calculated units of food prepared per hour are in a realistic order of magnitude, assuming a reasonable mix of different types of food being prepared and the overall emissions being dominated by those from grilling steaks, suggesting that the laboratory-derived emission factors for PN, $PM_1$, and organics are applicable to ambient measurements within an acceptable range of uncertainty.

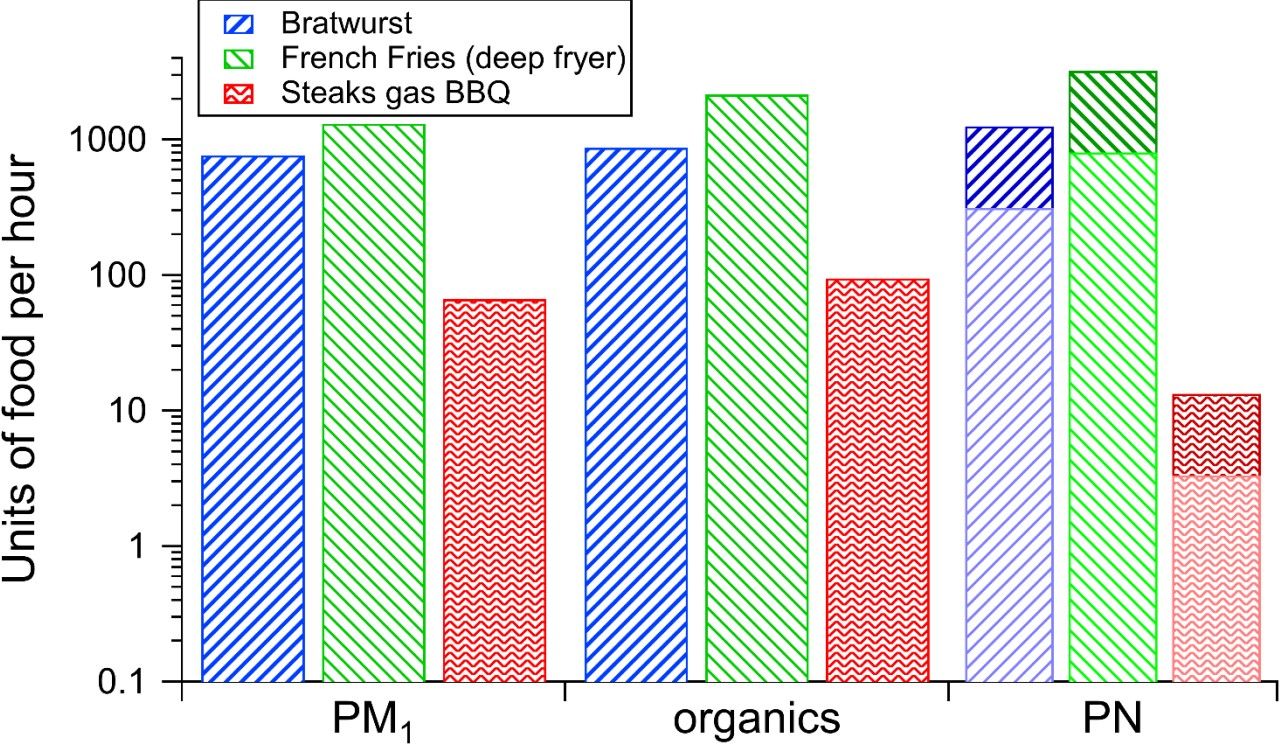

**Figure 14: Units of food that must be prepared per hour to generate the same concentrations (after background subtraction) as measured at the Christmas market in Bingen, calculated based on the emission factors for three different dishes and the local aerosol transport conditions. For each variable the corresponding COA fraction was calculated with $RIE_{COA} = 2.27$ and $CE_{COA} = 1$ and for PN a COA fraction range of 20% (light bar) and 80% (dark bar) was assumed.**

# 4 Conclusion

In a comprehensive laboratory study, various aspects of cooking emissions were investigated in real time with multiple instruments, including the chemical composition of $PM_1$ and particle size distributions, as well as emission dynamics and quantification of emissions through the calculation of emission factors. In addition, the influence of cooking activities on ambient aerosol was investigated at two German Christmas markets.

From the laboratory experiments, it was found that the measured particle number concentrations as well as several mass-based variables (PM, BC, PAH, organics) were strongly affected by the cooking activities. Measurements with the AMS suggest that the $PM_1$ fraction of the measured emissions contains a substantial fraction of vaporized and recondensed oil or fatty acids, as shown by comparing the mass spectra of the measured emissions with that of rapeseed oil, the used cooking oil. Therefore, we believe that particle formation and growth is to a large degree the result of oil vaporization or fat decomposition and recondensation of the emitted vapors.

By comparing the AMS-measured organics mass concentrations with the size-distribution-derived mass concentrations, we found that higher values of $RIE_{COA}$ (1.53 – 2.52) compared to the standard value of 1.4 are required to correctly determine the mass concentrations of cooking-related organic aerosols. These results confirm and extend the findings of previous studies. In conclusion, we recommend the use of different $RIE_{COA}$ values depending on the cooking oil, since it influences the $RIE_{COA}$: for cooking with rapeseed oil, an $RIE_{COA}$ of $2.17 \pm 0.48$ based on this study and the one by Reyes-Villegas et al. (2018), and for cooking with soybean oil, an $RIE_{COA}$ of $5.16 \pm 0.77$ based on the measurements by Katz et al. (2021).

In addition, to support the AMS data analysis of organic aerosol types, a new plot type is presented that provides an easy and quick way to check whether PMF has succeeded in separating different aerosol types using known markers, and also to identify and validate new markers, e.g. for real-time identification of aerosol types. By using data from multiple measurement campaigns, the variability of the mass spectra for individual aerosol types is taken into account and this provides the opportunity to evaluate how well the separation of aerosol types works based on the selected markers. Here we have identified and evaluated the ratio $f_{67}/f_{69} > 1$ as an additional COA marker. The presented examples show the importance of combining markers or indicators to achieve a robust separation from other aerosol types, such as for COA $f_{55}$ ($> 0.06$) and $f_{55}/f_{57}$ ($> 2$) for separation especially from HOA.

The relevant parameters that influence the amount of cooking emissions are the cooking temperature, the use of oil, the ingredients, and the activities during the cooking process. These are mostly dependent on the cooking method; therefore we observed similar results for dishes with similar cooking methods. A change in the concentrations of the relevant variables (PM, BC, PAH, organics) as well as in the particle size could be attributed to changes in the temperature of the food and the cookware as well as to different activities during the cooking. As the temperature increases, more substances vaporize and condense, resulting in higher emissions and larger particles. BC and PAH emissions were observed only at higher temperatures, e.g. towards the end of the cooking. Various activities lead to transient changes in concentration and particle size because they 1. facilitate the vaporization of substances, e.g., by stirring or tilting the pan, 2. increase the amount of vaporizable material, e.g., by cleaning the grill grid, or 3. suddenly release accumulated emissions, e.g., by opening the oven.

The ingredients used also have a strong influence on the aerosol composition. The emissions from boiled dishes differ from those of other dishes mainly due to the large absence of oil and fatty ingredients. Another example is the occurrence of sulfur-containing species in the emitted aerosol for dishes with fried onions.

In order to quantify the emissions, emission factors for all relevant variables were determined individually for all dishes. The highest emissions were released during the cooking of dishes on a gas and a charcoal grill due to the highest cooking temperatures, the burning of food residues from the grid, and, in the case of charcoal grilling, additional emissions from the burning of the charcoal itself. The emission levels from the cooking of stir-fried, deep-fried, and baked dishes were similar to each other as oil or

840 fatty ingredients were present. The cooking of boiled dishes resulted in the lowest emissions because no oil was used and no or only little amounts of fatty ingredients were available, limiting the amount of vaporizable substances. Furthermore, a comparison with other relevant indoor and outdoor emission sources showed that grilling one dish emits similar amounts of particles as driving 100 km in a car, and emissions from oil-based cooking, such as frying, are similar in magnitude to those from domestic wood burning over a comparable time period.

Average $PM_1$ concentrations during the opening hours of a Christmas market were found to be as high as 51 µg m$^{-3}$. Locally, visitors could be exposed to even higher concentrations, as shown by the BC concentrations measured with a portable aethalometer on the market, which were on average twice as high as those of the stationary measurements immediately downwind of the market. Although this is not a 24-hour average, these elevated concentrations show that events such as Christmas markets have a strong impact on local air quality.

This result, together with those from the laboratory measurements, shows that cooking activities contribute substantially to indoor and ambient aerosol. The amount of emissions is mainly determined by the cooking method, with barbecues being a particularly strong emission source.

*Author contribution.* JP and FD designed the measurements. JP performed the experiments, analyzed the MoLa data with support from FF, and drafted the paper with contributions from FD, FF, and SB.

*Competing interests.* The authors declare that they have no conflict of interest.

*Acknowledgements.* We thank Thomas Böttger and the mechanical workshop for technical support. The authors thank David Troglauer, Lasse Moormann, and Philipp Schuhmann for assistance with the laboratory measurements. We also thank the organizers of the Christmas markets for the opportunity to perform our measurements. We thank the Max Planck Institute for Chemistry for funding this work.

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
