# Peer review of "Particulate emissions from cooking: emission factors, emission dynamics, and mass spectrometric analysis for different cooking methods"

_EGUsphere, 2023_

## Author Comment (AC1)

**Referee report #1**

**What did I understand this paper to be about:**

This paper describes cooking experiments where a handful of different techniques and ingredients are used. The authors quantify total emissions for each dish, as well as emissions for individual phases/methods within the cooking of each dish, which are novel. There is considerable effort given to placing the experimental results in context of both emissions from other common sources as well as ambient PM measurements. The experiments appear to be well-done, with one major possible exception (which I note in detail) of whether or not temperature was measured.

→ Thank you for this general comment. We reply to the temperature measurement question further below.

**Big picture comments:**

-With due respect to the work—which appears to be well done, has novelty, and is largely presented well visually—the writing throughout this paper is poor. It is not publishable in its current form, in my opinion. In the beginning of my reading of the manuscript, I was attempting to do copy-editing, and line-by-line suggestions for grammatical changes, but I lost steam. It's a pervasive issue throughout the manuscript that needs a lot of revision. My best suggestion to the authors would be to recruit a few scientific colleagues (which would not have to be subject-matter experts) to specifically help with re-writing the manuscript. Improving the grammar of the manuscript and aiming for clear communication should be the primary goal of this possible collaboration.

→ We revised the whole manuscript with regard to the writing.

-Noted below in detailed comments for "Introduction," but there needs to be a reframing of what exactly is novel about this study. It is not clear from the abstract nor introduction why this is novel and takes a step further than what others have done.

→ To make the novel aspect of this study in comparison to previous studies clearer, we added a sentence to the abstract that frames our study within the existing work. See our reply for the introduction below.

-There are numerous statements and conclusions made in relation to the temperature of the prepared food/oil and cookware. Temperature has been shown to be an important variable in cooking emissions before, and this study identifies it as being key as well. And yet there is no measurement of it? This is a large issue to my mind. At the very least, it needs to be explicit that temperature was not measured (if indeed it wasnt?) and that the statements made about

temperatures are educated guesses. Or, if it was measured (again, not clear to me), then those data should be reported in some way.

→ Indeed, temperature was measured: In Section 2.1, where the laboratory study design and the experimental procedure is described, we stated that "the temperature of the cooked food and of the cookware was measured repeatedly with an IR thermometer (Fluke 568, Fluke Corporation, USA). During the baking experiments, the temperature inside the oven was monitored continuously using the same thermometer with a thermocouple". However, since this information apparently was not entirely clear, we now clarified this by adding the information that the "surface temperature of the food and the cookware" were measured with the IR thermometer, and state how often these measurements were performed. For the oven measurements we added the information that the "temperature of the air" was measured in the oven.

-There is a pervasive sense that the AMS PMF "COA" factor is simply volatilized oil. This is suggested by the simple mass spectral comparison using Pearson's R. However, the PM emissions factors of BBQ (where no rapeseed oil is mentioned with e.g., vegetable skewers, and very unfortunately is used for marinating steak—this would have been a nice opportunity to see what steak emissions are like in the absence of rapeseed oil (in the U.S. at least, few are marinating steak in oil like this…)) are going to largely not be associated with oil, and are orders of magnitude higher. There appears to almost be a "super-emitter" problem when comparing pan-prepared food vs. BBQ, which is shown by the steak vs. brats vs. fries calculations presented near the end. Some discussion of this super-emitter issue, especially when the emissions from BBQ are likely largely not rapeseed oil, deserves some attention, in my opinion.

→ We thank the reviewer for raising this important point! Indeed, to conclude from the correlations that cooking emissions are simply volatilized oil is likely to overstate the observations and oversimplify reality. We have revised the text (throughout the manuscript) to make it clear that not only rapeseed oil, but also other cooking oils and fatty components of the food are major contributors to cooking-related emissions and therefore significantly influence the associated mass spectra. In addition, we now state that this is consistent with the fact that such components can be readily emitted from the cookware, whereas peptides and carbohydrates are more likely to decompose into non-condensing or non-volatile products. The relevant text now reads (Section 3.1.1):

"Most of the spectra show a high degree of similarity to each other and to the spectrum of rapeseed oil (Pearson's $r > 0.94$), suggesting that the emissions are associated with oil, which might have vaporized and recondensed. Consistently, the mass spectra of the emissions from boiled dishes and steaks grilled on charcoal are less similar to those of the others: For the boiled dishes, no oil was used, and for the steaks, the mass spectrum is strongly influenced by the emissions from the charcoal itself. In addition, the correlations of the cooking mass spectra with those of different fatty acids (palmitic, stearic, oleic, and linoleic acid), all measured by AMS (Ulbrich et al., 2023, not shown in Fig. 2), show the highest similarity with that of oleic acid ($r = 0.85 - 0.94$), the main component of rapeseed oil and many other cooking oils. These observations suggest that a substantial fraction of cooking-related emissions are fatty acids, either from the used cooking oils or from components of the prepared food. This is consistent with the fact that oil components may vaporize and recondense, and fats may produce condensable fatty acids after decomposition. In contrast,

peptides and carbohydrates are more likely to decompose into products that either remain in the gas phase or do not vaporize under the cooking conditions."

In fact, the emissions from the mix of cooking methods at the Christmas market are dominated by the emissions from grilling steaks because, as discussed in Section 3.2, emissions increase sharply as the temperature of the cooking process increases. Due to the often very high temperatures involved in barbequing, it is not surprising that this cooking method produces exceptionally high emissions. We have added a statement to the revised manuscript to clarify this point. In addition, this point was already mentioned in the very last sentence of the conclusions.

**Detailed comments:**

**Abstract:**

L11: get rid of "both number and mass wise"

→ It was important to us to make clear that both, the mass of emitted aerosol, but also the number of particles are affected. We reworded this sentence: "… emissions, in terms of both number and mass, with often …"

L13: get rid of "and parameters" (not clear what this means)

→ What is meant is cooking activities like stirring or tilting, but also parameters like the temperature of the food and cookware. Since these external influences that affect the emissions are an important focus of our study, we want to keep this information; so, we reworded the sentence, omitting the word "parameters": "To investigate the characteristics of cooking emissions and what influences these emissions, …"

L13: "in THE form of…"

→ "in form of a measurement series" was completely removed.

L13: This whole sentence needs to be re-written. Grammatically un-sound.

→ As part of the revision of the writing of the whole manuscript, also this sentence was improved.

L13: "The emissions" would refer to everything—condensed and gas phase—so they way this is written is gas" coming at the end, as if that isnt a part of "emissions," or is auxiliary.

→ In the re-worded version of the sentence, it was taken into account that with "emissions" both particle phase and gas phase emissions are meant.

L17: "For six variables, we observed changes during the cooking:" — this is very poorly worded, suggest re-writing this sentence.

→ done

L18: "Organics mass concentrations" should be "organic aerosol mass concentration."

→ done

L20: "The emission dynamics of the above-mentioned variables" — this sentence is poorly-worded on a variety of levels.

→ In the re-worded version of the manuscript, this sentence was also improved.

L24: the last paragraph of the abstract is interesting, but again is very poor grammatically.

→ In the re-worded version of the manuscript, this paragraph was also improved.

General comment: I started to copy-edit this for grammar from the beginning of my reading, but am stopping now because there are too many instances to deal with in my role as a reviewer. I will make a "big picture comment" on this issue above, but this paper needs a lot of help in improving the writing before it can be published.

→ Thank you for this attempt to improve the language and grammar. After implementing the non-language related comments, we have revised the whole manuscript to improve the writing. To make the changes easier to follow, we have attached two different versions of the revised manuscript with the changes highlighted: One with only the non-language related comments included, and one final revised version with also the language improved.

**Introduction:**

-General comment: the literature on cooking is summed up fairly well, though I think one major omission that should be briefly mentioned is the extent to which cooking emissions may form secondary PM. This is one of the big open questions about cooking emissions and

associated PM, in my opinion. I realize this is not the focus of the manuscript, but here in the introduction when you are trying to sum up the spectrum of what is important with cooing as an emissions source this should be mentioned.

→ Thank you for this comment. We have added a paragraph to the introduction that addresses studies that examine secondary aerosol formation from cooking emissions and analyze potential precursor gases in the emissions.

-There is a disproportionate amount of time/space given towards summing up "the state of knowledge" in cooking emissions and their impacts. However, the end of the introduction does not, in my mind, really set up what is novel about this study in particular. I would suggest trying to reframe more specifically what is novel about your work, because it doesn't shine through in reading your introduction.

→ We added a paragraph between the summing up of the state of knowledge in cooking emissions and the description of our study, where we summarize the limitations of the previous studies and describe how our study is attempting to overcome them.

**Methods:**

-section 2.1: is temperature measured? If so, how? For this to really be a "systematic" study of cooking emissions, I expected to see temperature being measured. It's one of the key variables. I see no mention of what the ambient temperature of the experimental hall/kitchen is, nor, more importantly, what the temperature is of the food being cooked and/or cookware being used. This omission is striking given that temperature is mentioned within this paper as a key variable influencing emissions (e.g. L737: "The relevant parameters influencing the amount of cooking emissions are the cooking temperature, use of oil, ingredients, and activities during the cooking process." or e.g. L386: "An increase of BC and PAH mass concentrations was observed only for cooking methods operating at high temperatures like grilling or in the final phase of preparing stir-fried dishes." How is the following statement made in the absence of a temperature measurement? How is Figure 6 even constructed?). There is even mention of "an increase of the food and cookware temperature, as deduced from repeated temperature measurements" (L376), but no details of the measurement or actual quantification of temperature presented as far as I can tell?

→ The temperature of the food and cookware was measured repeatedly throughout the duration of each cooking session using an IR thermometer at selected locations on the food or cookware. During the oven experiments, the oven air temperature was measured continuously with a thermocouple. All of this was mentioned in the first version of the manuscript (lines 111-114). This information about the temperature measurements was slightly expanded (information that the IR thermometer measured the surface temperature and the thermocouple measured the air temperature in the oven; information about the frequency of the IR thermometer measurements). Due to the substantial heterogeneity of the temperature distribution throughout the food and the cookware, it is not feasible to associate individual emissions with specific temperatures at the (unknown) point of emission. Consequently, only

qualitative conclusions about the dependence of emissions and temperature can be drawn, as illustrated in Figure 7. This information was incorporated into the revised version of the manuscript. We also added information about the temperature in the experimental hall.

-Simply assuming CE=1 "because liquid" does not seem appropriate. You measure plenty of BC in ambient cooking emissions at the market, to illustrate this point, and are doing . I strongly suggest taking the approach of reporting the "response factor," similar to Katz et al., of RIE x CE. I understand that this equates to RIE when CE=1, but I think it is more honest to be clear that the response factor is the fundamental thing you are able to assess, as opposed to RIE since you aren't measuring CE itself.

→ We agree that choosing a CE value is a delicate task. In our laboratory experiments, we typically observed 100 times (range: 40 – 1000) larger organic aerosol emissions compared to BC emissions. Therefore, we conclude that the organic-containing cooking-related particles are massively dominated by organic material and are therefore most likely liquid. Therefore, we believe that we can assume a CE value of 1 and determine a relative ionization efficiency for the cooking related organics for the laboratory measurements as discussed in Section 3.1.4.

To make these assumptions clearer and to mention the limitations of these assumptions, we have modified the relevant text in Section 2.4.

**Results:**

-general comment: I strongly recommend changing the way that your axes are labeled in all of your plots—you have the format of "variable / units." This is something I rarely (maybe never?) see. Instead please change to the standard "variable (units)."

→ It is quite possible that the reviewer did not see this way of labeling the axes in our plots because it is rarely used. However, this way ("variable / unit") is the way recommended by SI and, for example, NIST, over the more commonly used "variable (unit)" way.

This is stated, for example, in Section 7.1 ("Value and numerical value of a quantity") in the "NIST Guide for the Use of the International System of Units (SI)"; NIST Special Publication 811:

"More formally, the value of quantity $A$ can be written as $A = \{A\}[A]$, where $\{A\}$ is the numerical value of $A$ when the value of $A$ is expressed in the unit $[A]$. The numerical value can therefore be written as $\{A\} = A / [A]$, which is a convenient form for use in figures and tables. Thus, to eliminate the possibility of misunderstanding, an axis of a graph or the heading of a column of a table can be labeled "$t/ºC$" instead of "$t$ (ºC)" or "Temperature (ºC)." Similarly, an axis or column heading can be labeled "$E/(V/m)$" instead of "$E$ (V/m)" or "Electric field strength (V/m)." …"

-Figure 2: these comparisons are are pinned to the spectra of rapeseed oil. But I don't see any discussion of why that is the choice of the reference spectra. Please note why this choice, and what you lose/gain by showing only this comparison. Also, you could easily show the correlation with another reference spectra—currently, this figure is duplicating the displayed information by a factor of two. I would suggest refining this figure considerably and/or moving it to SI. If anything, it would be more interesting to show this same figure with some canonical "COA" spectra, as opposed to rapeseed oil.

→ Figure 2 shows a comparison of the mass spectra of all cooking experiments with each other and additionally with the mass spectrum of the rapeseed oil used in the experiments. As discussed in the accompanying text, this figure is intended to show the similarity of all the emission spectra of all the experiments in which rapeseed oil was used and the similarity of all these spectra to that of rapeseed oil. The conclusion of this comparison is that fatty acids, like those from rapeseed oil, are likely to be a major component of the emissions from the respective cooking experiments.

It is not the case that rapeseed oil is the reference spectrum; rather, all spectra are compared to one another in this study. The choice of rapeseed oil for comparison is motivated by the fact that this was the oil used in the cooking experiments, as mentioned in the text.

We agree that a similar comparison with typical organic aerosol types, e.g. COA (cooking related organic aerosol), is also helpful to find general similarities. Therefore, we have included such a comparison in the Supplement (Figure S2) and referred to this figure in the main text.

We also agree that this figure duplicates the information presented. Unfortunately, it is not possible to merge Figure 2 and Figure S2 into a single figure because different groups of spectra are selected on the x- and y-axes in Figure S2.

We revised the text related to Figure 2 to clarify the points raised by the reviewer.

-L226: "therefore we assume that also during field measurements the detected cooking-related emissions mostly consisted of vaporized and re-condensed oil." This seems like a really big leap, given that a LOT of cooking is done with oil-temperature combinations where the smoke point is coming into play.

→ We agree with the reviewer that this conclusion may be too general with respect to the observations on which it is based. We have therefore revised the text to narrow the conclusion.

-Section 3.1.2: this reads more like a literature review instead of results. A lot of what you are describing could (and should) be presented visually.

→ Thank you for this important point. We have revised the entire section to focus first on the results of our study; and then to compare our results with those of other studies to draw more general conclusions. In addition, we have included the previous Figure S3 in the main text (new Figure 3) to visually present what is described in the text.

-Table 3: why is HOA not included here? Suggest inclusion. It should be made much more clear what from this table is from the literature vs. this study.

→ We have followed the reviewer's suggestion to include HOA in Table 3 and revised the associated text accordingly. We have also made it clearer which value in Table 3 is from this study. The fact that the other values are from the AMS mass spectra database is mentioned in the table caption.

Section 3.1.3: I'm confused why, after reading a page about m/z 60 and 73, why these are now absent or considered in any kind of rectangle plot here?

→ Thank you for this important comment, which has made us aware that we have not been clear enough about the use and meaning of the rectangle plots. We have therefore revised Section 3.1.3. to clarify that this type of plot is useful for identifying combinations of markers (rather than individual markers alone) to discriminate between different aerosol types. In addition, we have added a rectangle plot that includes the f60/f73 ratio to the supplement and mention it in the main text. Since in this case, the rectangle plot does not further improve the separation between BBOA and cooking-related organic aerosol compared to the f60/f73 ratio alone (which is already listed in Table 3), we chose not to additionally include this plot in the main text.

-Section 3.1.3: Given the abundance of experiments conducted, I am really surprised that this whole section—which is about relative fractions of 55 and 57—does not include a discussion on the variability of the actual molecular ion fragments, as e.g., "55" is a combination of C4H7+ and C3H5O+. Are these ratios always the same? This dataset seems rich is terms of helping us better understand what "55" means for cooking aerosol, perhaps even during different phases of cooking the same dish (speculating).

→ We are grateful for this suggestion. In response to this comment and a similar suggestion by the other reviewer, we have added a discussion of the ion fragments observed in the individual marker m/z signals to Section 3.1.3. Here, we focus on the discussion of the various ion families observed in the marker m/z signals, but also discuss the contribution of individual ions within these families. No clear distinction could be identified between the marker signal compositions observed across different cooking methods, with the exception of the enhanced contribution of oxygen-containing ions to m/z 57 and 69 in the emissions from grilling, in comparison to the other cooking methods, which is now also discussed in the main text.

Section 3.1.

-Figure 7: I applaud this figure and the work that went into it. But for legibility, please consider adding text in the main figure as to what the symbols mean. I dont think it's appropriate to make the reader have to refer to another figure to decipher this. I imagine you

can just move around the images an add extra text boxes, but if limited space is really the issue then consider replacing the quantities (e.g., "1.5-5x") with the activity (e.g., "Tilting pan") while keeping the color-coded order of magnitude indicators.

→ We revised Figure 7 (now Figure 8) according to the reviewer's suggestion and removed Figure S5 (which contained the meaning of the symbols) from the Supplement.

-Section 3.1.4 - Why is there only a single marker on Figs 3 and 4 for rapeseed oil? Were there not replicates? No variation at all? Your assumption that so much of your emissions are essentially rapeseed oil seems to not square with the fact that there is very little overlap of the "RO" marker and any of the boxes from either of these plots.

→ In response to this and the next comment, we revised the rectangle plots in the manuscript. The measurements of vaporized rapeseed oil were performed only once. We calculated the uncertainty of the corresponding marker values from the temporal variability of the mass spectra during the respective measurements. This uncertainty is now reflected by boxes for the rapeseed oil-related markers in the rectangle plots. In addition, we added the PMF results from both Christmas market measurements separately to the plots. The respective uncertainties were taken from repeated PMF analyses with different starting conditions ("seed" values).
We recalculated the data for all aerosol types where marker ratios were used (from – previously – first averaging the $f_{xy}$ values and then calculating the ratios to – now – first calculating the ratios and then averaging over all ratios). This resulted in slightly different positions and sizes of the corresponding boxes, but did not change the resulting conclusions. In fact, as the reviewer remarks, the RO marker is shifted to slightly larger f55 values compared to the results from the cooking experiments. We assume that this is due to the fact that while the cooking emissions or their mass spectra may be dominated by vaporized rapeseed oil / fatty acids, they likely also contain other components. We have clarified this in the text.

-Section 3.1.4 - There is very little discussion about how your experiments do not have much overlap with the "COA" box in either of these plots (figs. 3 or 4). Also, it seems like you are presenting the COA box from entries in the AMS MS database. I would strongly suggest adding your own COA spectrum from the market to this as well for completeness.

→ As suggested by the reviewer, we have expanded the discussion of possible reasons for the discrepancy between the observed marker values for the laboratory experiments and the PMF results from ambient aerosol. We have also added the marker values for our own COA spectra from the Christmas markets to the two figures.

-figure 9: pretty difficult to read. The background colors are pretty dark, and the light green markers barely stand out. One simple improvement would just be to make the markers black and not light green. I myself am not colorblind, but can imagine this presentation would be extra difficult for those who are. Worth verifying that the combinations of colors used in this plot are 'colorblind-friendly'

→ Thank you for this note. We have changed the green markers to black and increased the font size slightly to make it easier to read. In addition, we tested the plot for colorblindness-friendliness using a colorblindness simulator. As a result, we slightly changed one of the background colors. Now it should be completely readable with any type of color blindness.

-Figure 10: Strongly recommend moving BC and PAH EF results out of SI and into Figure 10. These figures can be made much smaller, without sacrificing the ability to read them, which will allow for the BC and PAH results to be shown in the main text. That "one unit" of barbecuing contributes a roughly equivalent amount of both BC and PAHs to the atmosphere as does "one unit" of traffic, is notable and worth putting in the main body.

-Figure 10: It becomes clear when sifting through the text what the units of "g" or "#" mean here, but it should be clear (or at least hinted at) within the Figure/figure legend itself. There needs to be some note that these are "Emissions per unit activity" or something along those lines, so that the reader understands that we are comparing e.g., a cooked dish to smoking two cigarettes to driving 100 km.

→ As recommended by the reviewer, we have moved the BC and PAH emissions comparisons to Figure 10 (now Figure 11) to provide these comparisons for all four variables. In addition, we have added the word "emissions" to the y-axis legend of all four panels and added "total emissions per unit activity" to the figure caption to help the reader better understand what is being shown without consulting the main text.

-Section 3.5.2 - Recommendation: take the calculation one step further—tell the reader how many bratwursts/orders of fries/steaks these mass ranges equate to using some e.g., average unit weight. This will 'make real' this calculation for the reader in a way that "80 kg bratwurst" does not.

→ Thank you for this very helpful suggestion. We converted the calculated masses of food needed to generate the observed emission concentrations into numbers of units of food. For this conversion, we assumed the mass of a single bratwurst to be 150 g, the mass of a schnitzel to be 180 g, and the mass of a unit of French fries to be 250 g, which is consistent with the average values from the laboratory cooking experiments (see Table S1). Figure 13 (now Figure 14) has been updated accordingly.

-Section 3.5.2 and Figure 12: the text alludes to the comparison being better after adjusting the response factor for COA, "the PM1 values align reasonably well with the one-to-one line." There are so many data points on this plot that visual perception of the improvement of the comparison is suspect. The improvement can easily be quantified by e.g., reporting the slope of the fit, and should be.

→ Thank you for raising this point. Indeed, the correlations depicted in this figure provide only a relatively weak indication that the RIE and CE values, derived from the laboratory experiments, result in a superior mass closure compared to the standard values. Consequently,

the text has been revised and the conclusion modified to be slightly less definitive. Furthermore, the text now includes the suggested information on the slopes of the fits.

-Section 3.5.2 - The following statement is made: "These calculated masses of food prepared per hour are all in a realistic order of magnitude (especially for the steak dish)," which implies that the combination of the laboratory emission factors for steak BBQ compares well/realistically to what is observed at the market. And yet, throughout the manuscript, there is a repeated assumption that the COA can largely be understood at volatilized oil (rapeseed oil specifically). However, barbecued steak is not cooked in oil and does not contain "oil;" most of the OA emissions from steak itself are presumably from volatilized fats, though emissions from flame/charcoal will also be a part of the mix from the market cooking. How do we square this?

→ In response to one of the general comments (see above), we have revised the conclusions and statements regarding the nature of the cooking emissions throughout the manuscript. We now conclude that the mass spectra of cooking emissions show clear signatures of fatty acids, suggesting that a substantial fraction of the cooking emissions are fatty acids, either from vaporized and recondensed oil (cooking oil or as natural part of the food) or from decomposition of fats in the food.
We agree that marinated steak contains residues of the marinating oil, but even non-marinated steak contains fatty components that may decompose under the influence of the very high temperatures during grilling and produce emissions with a similar mass spectral signature. The emission factors may be slightly different from those of marinated steak, but due to the high temperatures, we also expect strong emissions from grilling non-marinated steak or e.g. bratwurst in the same order of magnitude as in our measurements.
We have added a statement to Section 3.5.2 regarding the emissions from the Christmas market to make this point clear.

Referee report #2

**Summary**:

This paper presents a detailed description of particulate matter emissions from cooking in controlled laboratory experiments (e.g., boiling, baking, frying, grilling) and at two German Christmas markets. An AMS, OPCs, Aethalometer, and other instruments were used to describe the physical and chemical properties of the cooking emissions.

Laboratory and ambient observations were connected using a few methods, which I found interesting and novel. (1) The amount of food prepared at the German markets was estimated using laboratory-derived cooking emission factors and estimated ambient emissions of cooking organic aerosol. (2) The chemical composition of ambient cooking organic aerosol was compared to laboratory cooking emissions using introduced rectangle plots. (3) Laboratory-derived quantification parameters for cooking organic aerosol were used to improve agreement between the AMS and collocated instruments.

The laboratory results are presented with great detail – the authors show which cooking actions (e.g., stirring, tilting, flipping, etc.) cause the greatest emissions of PNC, PM1, BC, etc. They also present a few methods for comparing AMS spectra and identifying the signatures of cooking emissions using a few specific tracer m/z. The authors conclude that aerosolized oil is mainly responsible for cooking emissions due to spectral comparisons between heated oil and the cooking emissions.

→ We thank the reviewer for this general assessment.

**General comment:**

I agree with the other reviewer - the grammar requires improvement before publication. For that reason, I selected major revision. However, I believe the scientific methods of the paper are sound and thorough, and the results are important and novel. The figures and content are of decent quality and require only minor revision.

→ We completely revised the manuscript in order to improve the language. To make the changes easier to follow, we have attached two different versions of the revised manuscript with the changes highlighted: One with only the non-language related comments included, and one final revised version with also the language improved.

**Specific comments:**

Line 160: "Mobile laboratory" and "MoLa" are side by side in this sentence – probably a typo?

→ We have introduced the acronym "MoLa" in this sentence. To clarify this, we have now put the acronym in parentheses.

Line 157: Did the wind direction correspond with the mobile laboratory being downwind of the Christmas markets? It would be helpful to include this detail in the text.

→ Yes, in fact, the location of the mobile laboratory was chosen according to the dominant wind direction, and as a result, it was downwind of the Christmas markets most of the time. We have included this detail in the text.

Line 192:  How did you determine which spectra to compare to in the database? Were all spectra of a given type (e.g., OOA) utilized? It would be helpful to mention this briefly in the main text.

→ For each of the organic aerosol types listed in Table 2, all high-resolution mass spectra available in the database at the time of data analysis were used in this analysis. We included this information in the main text.

Section 3.1.3: Since you are using high-resolution AMS, why not mention the specific ion fragments in m/z 55 and 57 and the relative abundances of each?

→ We are grateful for this suggestion. In response to this comment and a similar suggestion by the other reviewer, we have added a discussion of the ion fragments observed in the individual marker m/z signals to Section 3.1.3. Here, we focus on the discussion of the various ion families observed in the marker m/z signals, but also discuss the contribution of individual ions within these families. No clear distinction could be identified between the marker signal compositions observed across different cooking methods, with the exception of the enhanced contribution of oxygen-containing ions to m/z 57 and 69 in the emissions from grilling, in comparison to the other cooking methods, which is now also discussed in the main text.

Figure 3 and 4: I like these figures and find they provide a nice way to differentiate the source of primary OA. It may be a helpful reference to include oleic acid and/or other oils on the plot if they are available. Additionally, it would be very helpful for the reader to include a brief definition of what the rectangles represent in the figure caption (e.g., they represent the standard deviation of PMF factors from the literature?).

→ We have included oleic acid, the major fatty acid in rapeseed oil, in these plots. Since the f55 fraction of oleic acid is significantly larger than f55 for all cooking-related aerosols, we have added a brief discussion of this fact to the text. In addition, we have added information to the figure captions of the two plots about what the rectangles represent (standard deviation of the available data).

Section 3.1.4: A very nice and thorough summary of the quantification is presented here. One area for improvement is the discussion of CE. I think a reference is needed in line 322 to justify using a CE of 1. Or, as the other reviewer suggested, combining RIE and CE as one response factor may be justified since CE was not measured.

→ In laboratory measurements (and also in field studies) it has been shown that liquid particles, including liquid organics, have a CE of 1. Therefore, and because in our laboratory studies only a negligible fraction of the aerosol mass is black carbon or inorganic material, we decided to use CE=1 for our analysis. As suggested, we have added a reference to the text justifying the use of CE=1.

Section 3.1.4: After seeing the scatter in Figure 12 I am curious about the comparison between the AMS and other instruments during laboratory experiments. Did you see a high correlation for those experiments? It would be helpful to see the scatter plots (in the SI maybe?) showing how you derived the response factors. I think it's important to show quantitatively that there was good agreement between the trends in signal reported by the collocated instruments before utilizing them to calculate a response factor.

→ As suggested by the reviewer, we have included several scatter plots, which were used to determine RIE values from the laboratory experiments, in the Supplementary Information (Figure S4). For each cooking type for which RIE values were determined, an example scatter plot is included in this figure, selected to cover the observed range of correlation coefficients. A reference to this figure has been added to Section 3.1.4.

Figure 6: I think there should be some explanation as to where the temperature data is coming from. In the methods you write that there were temperature measurements, but do not report them in the results section.

→ We did not plot the temperature data because it was measured repeatedly, but not on a regular and high frequency basis. In addition, the temperature distribution across the food and the cookware is strongly inhomogeneous and no information on the temperature of the actual location of the strongest emissions is available. Therefore, the temperature data cannot be used to generate correlation plots or anything like that. For this reason, the temperature dependence of the emissions is shown only schematically in Figure 6 (now Figure 7). To make this clearer, we have slightly modified the main text in several places. In addition, we have repeated in Section 3.2 the information from the methods section that the temperature measurements were made by repeated manual temperature measurements with an IR camera.

Figure 7: Please denote what the icons mean.

→ The meaning of the icons has been added to the revised version of the figure.

Line 423: You mention that the size distributions for the different methods were "partially significant." If you aren't referring to statistically significant, please use different language.

→ What was meant was "statistically significant". We have revised the text accordingly to make this clearer.

Line 540-545: I personally feel like too much text is used to describe the methods of the other study. Your summary which begins on line 546 suffices.

→ We agree that we have provided a lot of detailed information here. Since we believe that some of this information is necessary to understand the large differences from the other study to our results, we have greatly shortened this description instead of deleting it completely.

Line 552: The use of the labels "traffic" and "biomass burning" are somewhat misleading. To me, "traffic" implies multiple cars, but you are only including one car driving 100 km if I'm not mistaken? And again, to me "biomass burning" implies burning on a much larger scale that what you are using. Perhaps consider switching to "car" and "wood home heating" or something more clear, or indicate exactly what you mean in the figure caption.

→ We agree that these two labels are somewhat misleading. Along the lines suggested by the reviewer, we changed them to "wood home heating" and "driving a car" in the main text and in the related figure, as well as in the Supplement.

Figure 12: What do you suspect is causing the scatter in this plot? How was your agreement during normal ambient sampling (not during the cooking market)? Was there a similar amount of scatter?

→ The relatively strong scatter in Figure 12 (now Figure 13) is probably mainly caused by the very close proximity of the measurement location to the sources. As shown in Figures S6 and S7 in the Supplement, all pollutant concentrations show a very strong variability during the opening hours of the Christmas market. When translating this into average concentrations for individual time intervals (as used for the correlation plots in Figure 12), this can lead to increased scatter of the data, e.g., due to different coverage of the interval for the actual measurement as in the AMS, which alternately measures background and aerosol concentrations. For ambient measurements without nearby strong sources, the scatter in such correlation plots is much smaller and dominated by the measurement uncertainties of the instruments involved. We have added the reason for the large scatter in the main text.

Line 702: How did you determine 20 to 80%? This seems arbitrary… Consider presenting only the PM1 and organics results since you have the PMF factor for support.

→ Thank you for this comment. In fact, the 20 to 80% range has been arbitrarily chosen. We simply assumed that the "true" COA fraction for PN would be somewhere in that range. To

make this clearer, we have rephrased the sentence. As shown in Figure 14, where we wanted to show that these emission factors result in reasonable amounts of food having to be cooked to explain the observed concentrations, this range does not fundamentally affect the conclusion.

Figures S10: The enhancement in Chl is interesting. Why not report the other inorganic ion families from the AMS? Where do you suspect the Chl is from? It would be really interesting to see if NO3 was also emitted during these time periods.

→ The enhancement in particulate chloride during the Christmas market opening hours is probably due to emissions from wood burning on the markets. We have added this information to the main text. Other variables, such as other inorganic ion families from the AMS, were not shown in these two figures (now Figures S8 and S9) because they did not show enhanced values during the Christmas market opening hours. This selection is explained in the main text (first sentence of Section 3.5).

Figure S11 and S12: During the Christmas markets there is a clear enhancement in concentration relative to the nighttime and mornings. Do you have data from the same time periods on days the market wasn't happening? How do you know that some of the signal you are attributing to the market isn't a daily occurrence during those time periods? Some controls would provide more confidence.

→ We agree that additional measurements on days when there were no Christmas markets would provide information about regular daily patterns and would make the calculation of the Christmas market-related contributions more reliable. Unfortunately, we were only able to measure at the market locations for the duration of the Christmas market and do not have such comparative measurements. However, the location of the market is sufficiently removed from major roads and also from areas of strong activities during "normal" days (when there is no market happening), so we are quite confident that the concentration enhancements observed during the opening hours are predominantly caused by emissions from the Christmas market and related activities.

---

## Referee Report (RR1)

The authors did a good job addressing the comments from myself and the other reviewer. Two important notes should be addressed from my initial review before publication.

(1) In the original paper, the authors claimed a difference was 'partially significant,' and I asked for clarification. They corrected the sentence to say, 'partially statistically significant' (Line 499-500). To my knowledge, there is no such thing as 'partially statistically significant.' Any 'significant' claim should be backed by a specific significance test (e.g., the Student's T-Test) and have a corresponding p-value. If the resulting p-value is lower than the accepted p-value for rejecting the null hypothesis (e.g., $p < 0.05$), the authors can claim that a difference is significant, but there is no grey area with these tests. The result is either significant or it is not. Which significance test and p-value were used here?

The authors use the word 'significant' elsewhere in the text (Lines 252, 299, 345, 381, 383, 419, 427, 637, 701, 738). I believe that for many of these instances, they do not mean statistically significant, and the word should be replaced with e.g., 'substantially', 'notable', 'considerably', or 'to a marked extent'. Where the authors do mean statistically significant, they should denote the significance test used and the result (e.g., $p < 0.05$).

(2) Some grammar, especially in the introduction, requires further attention. Below are my line-by-line suggestions for improving the grammar and context for some of the claims.

Line 34-35: Change 'People tend to spend an increasing proportion of their time indoors, particularly in developed countries with about 90%, and are therefore exposed to indoor aerosol and its pollutants for long periods of time.'

Suggestion: 'People, especially in developed countries, spend a large portion of their time indoors (~90%), and are therefore exposed to indoor aerosol and other pollutants for long periods of time.'

Line 36-37: Change 'resulting health effects' to e.g., 'possible health effects of aerosol exposure'. Also, in line 36, when you write 'these pollutants', please specify what you mean. Is that aerosols? Or other pollutants which aren't mentioned?

Line 40: Change to 'Indoor aerosol composition is influenced by atmospheric infiltration, as well as multiple...'

Line 40 – 45: While evaporation/condensation could be a source of indoor particle mass, I think it is important to note that this an extremely minor source. The way it is currently worded as 'Aerosols can be generated by the evaporation...' also implies that evaporation is a source of new particle formation indoors, which I haven't seen, and is not supported by the Abbatt and Wang reference used in the paragraph. The Abbatt and Wang reference compares indoor surfaces and aerosol

particles as potential surfaces for SVOCs to condense, and indoor surfaces are far more important than indoor aerosols. Major indoor aerosol sources are infiltration and combustion (including cooking), and I think that the review of these sources in line 40-45 should reflect that. I provided recommendations to edit the paragraph below.

Change 'Aerosols can be generated by the evaporation of substances…' to 'Though a relatively minor source, evaporation and subsequent condensation of substances from furnishings, building materials, and consumer products, can contribute to indoor aerosol mass.' In line 45, change 'strong indoor emissions' to 'high indoor emissions.'

Line 51: delete 'through the stronger emissions'

Line 53: delete 'such ones with'

Line 91: Original sentence: 'The analysis of cooking emissions is challenging due to the high complexity of the emitted substance mixture, as well as the high emission dynamics with strong concentration variability during cooking.'

Suggestion: 'The analysis of cooking emissions is challenging due to the complexity of the emitted mixture, as well as the emission dynamics and concentration variability during cooking.'

Lin 598: Change 'That oil-based cooking (e.g. deep-frying and stir-frying) results in higher particle number concentrations compared to water-based cooking (boiling and steaming) has also been observed by See and Balasubra…'

Suggestion: 'Oil-based cooking (e.g. deep-frying and stir-frying) causing higher particle number concentrations compared to water-based cooking (boiling and steaming) has also been observed by See and Balasubra…'

Line 626: Include the references used in figure 11 here and in the figure caption instead of just saying 'from the literature'

Line 707-708: Change both instances of 'the one' to 'the fraction'

---

## Author Response (AR2)

**Reply to the reviewer comments on the revised version.**

Reviewer #1:

Reviewer #1 did not provide any comments and suggested publication as is.

Reviewer #2:

Comments on revised Pikmann et al.

The authors did a good job addressing the comments from myself and the other reviewer. Two important notes should be addressed from my initial review before publication.

(1) In the original paper, the authors claimed a difference was 'partially significant,' and I asked for clarification. They corrected the sentence to say, 'partially statistically significant' (Line 499-500). To my knowledge, there is no such thing as 'partially statistically significant.' Any 'significant' claim should be backed by a specific significance test (e.g., the Student's T-Test) and have a corresponding p-value. If the resulting p-value is lower than the accepted p-value for rejecting the null hypothesis (e.g., $p < 0.05$), the authors can claim that a difference is significant, but there is no grey area with these tests. The result is either significant or it is not. Which significance test and p- value were used here?

*Reply: Reviewer #2 is completely right. Of course, there is nothing like "partially significant" and this is not what we actually meant with the sentence. The sentence was unfortunately poorly worded from our side. Therefore, we revised the sentence:*

*"Therefore, the observed differences between the distributions for the different cooking methods were partially statistically significant."*

*Into what we really meant:*

*"Therefore, several of the observed differences between the distributions for the different cooking methods were statistically significant."*

*To determine significance, we did not use a dedicated significance test. We claim differences to be significant if the respective values disagree, taking their ranges of uncertainty (determined by, e.g., one standard deviation) into account.*

The authors use the word 'significant' elsewhere in the text (Lines 252, 299, 345, 381, 383, 419, 427, 637, 701, 738). I believe that for many of these instances, they do not mean statistically significant, and the word should be replaced with e.g., 'substantially', 'notable', 'considerably', or 'to a marked extent'. Where the authors do mean statistically significant, they should denote the significance test used and the result (e.g., $p < 0.05$).

*Reply: Thank you for this important hint. We revisited all instances of "significant" or "significantly", which were mentioned by the reviewer. In lines 299, 345, 381, and 383 the word "significant" was actually used in its statistical meaning and we kept it as it was. In lines 252, 419, 427, 637, 701, and*

*738 it was not used in its statistical meaning and we replaced it by "substantial" or "substantially" in the revised version.*

(2) Some grammar, especially in the introduction, requires further attention. Below are my line-by-line suggestions for improving the grammar and context for some of the claims.

Line 34-35: Change 'People tend to spend an increasing proportion of their time indoors, particularly in developed countries with about 90%, and are therefore exposed to indoor aerosol and its pollutants for long periods of time.'

Suggestion: 'People, especially in developed countries, spend a large portion of their time indoors (~90%), and are therefore exposed to indoor aerosol and other pollutants for long periods of time.' Line 36-37: Change 'resulting health effects' to e.g., 'possible health effects of aerosol exposure'. Also, in line 36, when you write 'these pollutants', please specify what you mean. Is that aerosols? Or other pollutants which aren't mentioned?

*Reply: We adopted the reviewer's suggestion.*

Line 40: Change to 'Indoor aerosol composition is influenced by atmospheric infiltration, as well as multiple…'

*Reply: We adopted the reviewer's suggestion.*

Line 40 – 45: While evaporation/condensation could be a source of indoor particle mass, I think it is important to note that this an extremely minor source. The way it is currently worded as 'Aerosols can be generated by the evaporation…' also implies that evaporation is a source of new particle formation indoors, which I haven't seen, and is not supported by the Abbatt and Wang reference used in the paragraph. The Abbatt and Wang reference compares indoor surfaces and aerosol particles as potential surfaces for SVOCs to condense, and indoor surfaces are far more important than indoor aerosols. Major indoor aerosol sources are infiltration and combustion (including cooking), and I think that the review of these sources in line 40-45 should reflect that. I provided recommendations to edit the paragraph below.

Change 'Aerosols can be generated by the evaporation of substances…' to 'Though a relatively minor source, evaporation and subsequent condensation of substances from furnishings, building materials, and consumer products, can contribute to indoor aerosol mass.' In line 45, change 'strong indoor emissions' to 'high indoor emissions.'

*Reply: We adopted the reviewer's suggestion.*

Line 51: delete 'through the stronger emissions'

*Reply: We deleted "through the stronger emissions" as suggested by the reviewer.*

Line 53: delete 'such ones with'

*Reply: We deleted "such ones with" as suggested by the reviewer.*

Line 91: Original sentence: 'The analysis of cooking emissions is challenging due to the high complexity of the emitted substance mixture, as well as the high emission dynamics with strong concentration variability during cooking.'

Suggestion: 'The analysis of cooking emissions is challenging due to the complexity of the emitted mixture, as well as the emission dynamics and concentration variability during cooking.'

*Reply: We adopted the reviewer's suggestion.*

Lin 598: Change 'That oil-based cooking (e.g. deep-frying and stir-frying) results in higher particle number concentrations compared to water-based cooking (boiling and steaming) has also been observed by See and Balasubra…'

Suggestion: 'Oil-based cooking (e.g. deep-frying and stir-frying) causing higher particle number concentrations compared to water-based cooking (boiling and steaming) has also been observed by See and Balasubra…'

*Reply: We adopted the reviewer's suggestion.*

Line 626: Include the references used in figure 11 here and in the figure caption instead of just saying 'from the literature'

*Reply: We understand that all relevant information should be in the main text. However, the bars of the pollutant sources, which were used for comparison, were determined from multiple values from*

*multiple studies in the literature. No individual values from these references were included in Figure 11. Since the literature search resulted in 26 values from 26 publications, we decided not to include all these references in the main text and in the figure caption but to summarize them in the supplement. Including 26 references in the main text (and in the figure caption) seems us to be inappropriate and we prefer to refer to the supplement for this in-detail information.*

*We referenced Table S7 that includes all 26 references in the figure caption to make clear that the references are listed in the supplement.*

*Please let us know if you insist that we include this whole list of references in the main text. Then we will add it to the text.*

Line 707-708: Change both instances of 'the one' to 'the fraction'

*Reply: We changed both instances of "the one" to "the fraction", as suggested by the reviewer.*

---

## Author Response (AR3)

Reply to the Editor's comments:

Thank you to the editor for suggesting these language improvements. We have implemented all suggested improvements in the new revised version.

Minor editorial comments:
*Abstract, 19: "For six variables we found evidence of emissions from cooking"; suggestion to revise as follows: "We found influence of cooking emissions on six variables". Additionally, suggestion to add out of how many variables, e.g., influence on six of x variables.

→ We have revised the text according the the editor's suggestion.

*Abstract, lines 19-20: "particle number concentration of smaller (particle diameter dp > 20 5 nm) and larger particles (dp > 250 nm)"; suggestion to revise as follows: "number concentration of smaller (particle diameter dp > 20 5 nm) and larger (dp > 250 nm) particles"

→ We have revised the text according to the editor's suggestion.

Figure 1: It is unclear why only HEPA is defined in the Figure 1 caption. It seems that some of the other abbreviations are spelled out in the text but not all of them. Suggestion to define all abbreviations that are not spelled out in manuscript or SI and add text that other abbreviations are defined in manuscript or SI.

→ We have checked Figure 1 for undefined abbreviations. The only abbreviation that was not defined in the Supplemental Information (Table S2) yet was NOx. We added this definition to Table S2 (which is referenced in the figure caption of Figure 1), where most of the abbreviations are defined.